# Curcumin in Cancer and Inflammation: An In-Depth Exploration of Molecular Interactions, Therapeutic Potentials, and the Role in Disease Management

**DOI:** 10.3390/ijms25052911

**Published:** 2024-03-02

**Authors:** Dong-Oh Moon

**Affiliations:** Department of Biology Education, Daegu University, 201, Daegudae-ro, Gyeongsan-si 38453, Gyeongsangbuk-do, Republic of Korea; domoon@daegu.ac.kr; Tel.: +82-53-850-6992

**Keywords:** curcumin, cancer, inflammation, docking

## Abstract

This paper delves into the diverse and significant roles of curcumin, a polyphenolic compound from the Curcuma longa plant, in the context of cancer and inflammatory diseases. Distinguished by its unique molecular structure, curcumin exhibits potent biological activities including anti-inflammatory, antioxidant, and potential anticancer effects. The research comprehensively investigates curcumin’s molecular interactions with key proteins involved in cancer progression and the inflammatory response, primarily through molecular docking studies. In cancer, curcumin’s effectiveness is determined by examining its interaction with pivotal proteins like CDK2, CK2α, GSK3β, DYRK2, and EGFR, among others. These interactions suggest curcumin’s potential role in impeding cancer cell proliferation and survival. Additionally, the paper highlights curcumin’s impact on inflammation by examining its influence on proteins such as COX-2, CRP, PDE4, and MD-2, which are central to the inflammatory pathway. In vitro and clinical studies are extensively reviewed, shedding light on curcumin’s binding mechanisms, pharmacological impacts, and therapeutic application in various cancers and inflammatory conditions. These studies are pivotal in understanding curcumin’s functionality and its potential as a therapeutic agent. Conclusively, this review emphasizes the therapeutic promise of curcumin in treating a wide range of health issues, attributed to its complex chemistry and broad pharmacological properties. The research points towards curcumin’s growing importance as a multi-faceted natural compound in the medical and scientific community.

## 1. Introduction

Curcumin, the vibrant yellow compound found in the root of the Curcuma longa plant, stands out as the primary curcuminoid in turmeric, belonging to the ginger family. Its molecular identity is characterized by unique polyphenolic elements, specifically two feruloyl groups linked via a methylene chain. This molecular configuration plays a crucial role in its biological and chemical properties [1,2].

For generations, curcumin has been a key component in the traditional medicinal practices of various Asian cultures. Its use in Ayurvedic and traditional Chinese medicine, along with other herbal systems, has been extensive. Curcumin has been applied to treat a wide range of ailments, from inflammation and pain to more specific conditions such as jaundice, menstrual issues, bleeding, dental pain, bruises, and cardiac pain [3].

In the realm of contemporary science, curcumin has sparked considerable interest due to its potential health benefits. Studies have delved into its effectiveness against chronic illnesses like cancer, Alzheimer’s disease, heart diseases, and inflammatory conditions [2,4,5]. This interest is fueled by its properties as an antioxidant and an anti-inflammatory, and its possible role in cancer prevention. Scientists are examining how curcumin influences various cellular processes by interacting with multiple signaling molecules, including growth factors, cytokines, and the genes involved in cell life cycle and division.

Despite its promising aspects, curcumin’s practical application in medicine is hindered by its limited bioavailability. This is attributed to its low solubility in water and rapid breakdown in the body. To overcome these barriers, the research is focused on developing innovative delivery mechanisms such as nanotechnology-based systems, liposomal encapsulation, and phytosomal formulations, aiming to enhance its absorption and efficacy [6,7].

Furthermore, ongoing research endeavors are dedicated to unraveling the precise molecular mechanisms through which curcumin exerts its effects. Although its influence on various molecular targets has been established, the detailed pathways of its action remain an area of active investigation. This understanding is vital for the development of curcumin-based therapeutic strategies and their integration into conventional medical treatments.

In summary, curcumin, with its deep-rooted history in traditional healing practices and its promising prospects in modern medical research, continues to be an area of keen scientific focus. The complexity of its chemistry, the breadth of its pharmacological actions, and the challenges related to its bioavailability present intriguing avenues for ongoing and future studies. As research progresses to elucidate its molecular actions and refine its delivery methods, curcumin stands as a potential key player in the treatment of a diverse spectrum of health conditions.

## 2. Curcumin’s Molecular Interactions and Target Proteins

Molecular docking, an essential technique in computational biology and pharmaceutical development, is key to deciphering how molecules interact, especially in the context of drug–target engagement [8,9]. This method predicts the manner in which a small molecule, like a pharmaceutical agent, binds to a specific protein. Such interactions are critical as they can identify prospective binding locations, gauge the drug’s affinity to the target protein, and determine its subsequent biological effects.

Focusing on curcumin, a polyphenolic substance sourced from turmeric, molecular docking becomes increasingly significant. Celebrated for its broad therapeutic range, including anti-inflammatory, antioxidant, and potential anticancer abilities, curcumin’s exact molecular action has been extensively explored in scientific studies [10,11].

Through the lens of molecular docking, researchers have been able to create simulations that illustrate how curcumin interacts with different proteins [12]. This is vital in pinpointing the exact locations on the proteins where curcumin attaches, which is integral to understanding its role in cellular mechanisms. In cancer research, for example, molecular docking provides insights into how curcumin binds with proteins that are essential in cancer cell growth. This binding can potentially suppress tumor growth, initiate cell death, or block the spread of cancer cells. Additionally, molecular docking has played a crucial role in investigating curcumin’s interactions with proteins involved in inflammatory and neurodegenerative diseases. In the case of inflammation, such studies can reveal how curcumin may inhibit critical enzymes and cytokines in inflammatory pathways. For neurodegenerative conditions like Alzheimer’s and Parkinson’s, docking studies are instrumental in revealing how curcumin might interact with amyloid plaques or tau proteins, thereby highlighting its potential neuroprotective properties.

A challenge in the field of curcumin research is enhancing its bioavailability [13,14]. Molecular docking is not just pivotal in understanding curcumin’s natural interactions but also assists in the design of curcumin derivatives or analogs with improved bioavailability and efficacy. By altering curcumin’s structure based on the docking outcomes, researchers can develop more effective and absorbable forms of the compound.

In summary, molecular docking acts as a crucial bridge linking the traditional use of curcumin with its contemporary therapeutic applications. This deepens our understanding of how curcumin interacts at a molecular level, aiding in the development of targeted treatments for various medical conditions. As the research advances, molecular docking continues to be an invaluable tool in unlocking the full therapeutic potential of curcumin, potentially leading to its more effective application in medical treatments.

## 3. Curcumin Target Proteins in Cancer

### 3.1. Cyclin-Dependent Kinase 2 (CDK2)

CDK2 is a crucial protein in cancer biology, primarily involved in the regulation of the cell cycle. It plays a significant role in the G1 and S phases of the cell cycle, where it helps control cellular proliferation [15,16]. CDK2, when combined with its regulatory partners, cyclins E and A, drives the cell cycle progression. It does this by phosphorylating key substrates that are necessary for DNA replication and the initiation of mitosis [17].

In cancer, the dysregulation of CDK2 is often observed, contributing to the unchecked growth of cancer cells [18]. The overexpression or hyperactivation of CDK2 leads to an accelerated cell cycle, enabling a rapid cell division that is characteristic of tumor growth. Furthermore, CDK2’s interaction with other cell cycle regulators can make cancer cells resistant to standard therapies, complicating treatment. Inhibiting CDK2 activity has emerged as a potential therapeutic strategy in cancer treatment. By targeting CDK2, it is possible to halt the proliferation of cancer cells, inducing cell cycle arrest, potentially leading to cancer cell death [19,20]. This makes CDK2 an attractive target for anticancer drug development, with several inhibitors being explored for their potential to disrupt its activity and thereby control the progression of various cancers.

In the study conducted by Riyadi Sumirtanurdin and colleagues, the interaction between curcumin (C_21_H_20_O_6_), its derivatives (referred to as kurkumod), and CDK2 was explored. The research was centered on analyzing how curcumin and its derivatives dock with CDK2 [21]. By using the CDK2 receptor (with the PDB ID: 1KE6) complexed with an antagonist ligand, the study aimed to elucidate the intricacies of this interaction, particularly focusing on the ligand’s binding at the active site. The use of high-resolution data was critical in accurately mapping the complex’s structure, which is essential for precise molecular docking. The research’s key findings included parameters like Gibbs free energy, inhibition constants, the types of bonds that were formed, and the number of clusters observed. Of particular interest were kurkumod 23 (C_17_H_22_N_3_O_3_S_1_) and 24 (C_17_H_22_N_3_O_3_S_1_), which demonstrated potential as CDK2 inhibitors. This potential was attributed to their lower Gibbs free energy and higher stability in cluster numbers, suggesting a stronger and more stable affinity for CDK2. Furthermore, these kurkumod derivatives were noted to form significant hydrogen bonds with amino acids such as Lys33, Glu81, and Leu83 on the CDK2 receptor. These interactions are similar to the binding dynamics observed between ATP and CDK2, further underscoring the potential efficacy of kurkumod 23 and 24 as CDK2 inhibitors. The functions of CDK2 and its inhibition by kurkumod 23 are depicted in Figure 1.

### 3.2. Casein Kinase 2 (CK2) Alpha

The serine/threonine-specific protein kinase CK2 alpha, also recognized as casein kinase 2 or CK2α, is notably influential in cancer biology. CK2α is primarily involved in fostering cell proliferation and enhancing cell survival [22,23]. It achieves this by phosphorylating numerous substrates that play a role in the pathways responsible for cell growth, thereby contributing to the rampant proliferation typical of cancer cells.

CK2α also exhibits properties that inhibit apoptosis [24]. It achieves this by phosphorylating and activating PI3K/AKT, which is crucial in the apoptotic pathway, thus blocking programmed cell death, which is a vital process for removing malignant cells. The PI3K/AKT pathway is critical for various cellular functions, including cell growth, survival, and cancer processes [25,26]. Activation occurs through growth factor binding to cell receptors, initiating a cascade involving PI3K and AKT, along with influencing downstream proteins [27]. PTEN opposes this pathway, but CK2 activates it by interacting with AKT and phosphorylating PTEN, reducing its activity [28,29]. The NF-κB pathway, activated by stimuli like cytokines, involves the IκB kinase (IKK) complex and influences inflammatory responses and cancer progression [30,31]. CK2α affects this pathway by phosphorylating IκBα, promoting the release of NF-κB transcription factors [32]. The JAK/STAT pathway transmits signals from receptors to the nucleus, regulated by SOCS proteins [33]. This pathway, influenced by CK2, plays a role in immune system development and various cellular events [34]. CK2 affects the phosphorylation levels of STAT3 and other components, impacting cancer cell survival [35]. CK2’s direct phosphorylation of JAK2 and its role in STAT3 activation highlight its importance in this signaling pathway.

CK2α’s extensive involvement in cancer makes it a potential target for therapeutic intervention. Strategies to inhibit CK2α could help in controlling tumor growth and enhancing the effectiveness of apoptosis in cancer cells. Furthermore, CK2α is linked to the development of resistance to cancer treatments, with its ability to activate survival pathways and hinder apoptosis potentially reducing the efficacy of chemotherapeutic drugs.

In research conducted by Giorgio Cozza and colleagues, the inhibitory effects of curcumin and its derivative, ferulic acid (C_10_H_10_O_4_), on CK2α were examined [36]. Curcumin exhibited an IC50 of 2.38 ± 0.15 μM against CK2α, whereas ferulic acid showed a more potent effect, with an IC50 of 0.84 ± 0.10 μM (Ki = 0.41). The study revealed that ferulic acid binds to CK2α, positioning its guaiacol group towards the hinge region and its carboxylic function in a positively charged area. This results in interactions with Lys68’s side-chain nitrogen, Asp175’s main-chain nitrogen, and a conserved water molecule, W1. Additionally, ferulic acid’s guaiacol hydroxyl forms a hydrogen bond with the main-chain nitrogen of Val116, and its oxygen forms a water-mediated hydrogen bond with Val116’s main-chain carbonyl, indicating a significant inhibitory interaction with CK2α. Figure 1 illustrates the roles of CK2α and how it is inhibited by ferulic acid.

**Figure 1 ijms-25-02911-f001:**
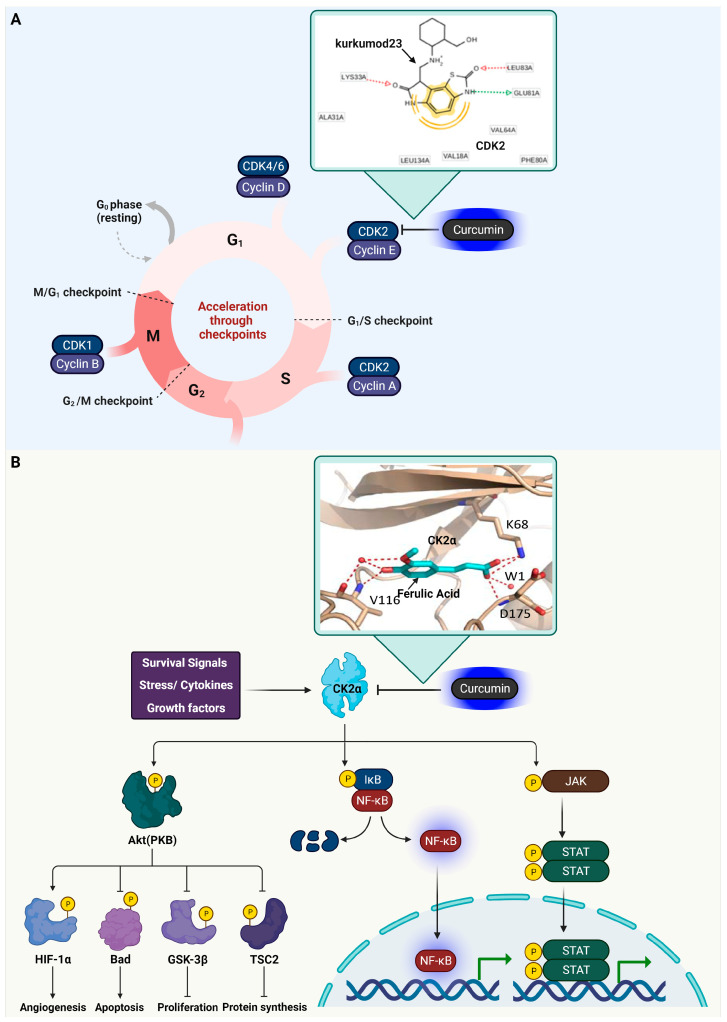
Interaction of Curcumin with CDK2 and CK2α. (**A**) The diagram illustrates the cell cycle phases, with a focus on the G1/S transition regulated by CDK2 complexes. Inset: Molecular docking of curcumin at the active site of CDK2 (PDB ID: 1KE6), suggesting potential inhibition. Key amino acids involved in hydrogen bonding with the kurkumod23 of curcumin derivative are Lys33, Glu81, and Leu83, which are crucial for the binding efficacy, similar to ATP’s interaction with CDK2. The docking image was cited from Reference [21]. (**B**) CK2α plays a regulatory role in several signaling pathways that are crucial for the development and function of immune cells. These pathways include (1) the PI3K/AKT pathway, (2) the NF-κB signaling cascade, and (3) the JAK/STAT signaling mechanism. Inset: Ferulic acid exhibits a notable inhibitory interaction with CK2α, where its guaiacol hydroxyl group forms a hydrogen bond with the main-chain nitrogen of Val116. Additionally, its oxygen creates a water-mediated hydrogen bond with the main-chain carbonyl of Val116. The corresponding docking image is cited from Reference [36]. Green arrows indicate gene expression. Yellow circles with ‘P’ represent phosphorylation. Cartoon in Figure 1 was created with BioRender.com (https://app.biorender.com, accessed on 10 January 2024).

### 3.3. Glycogen Synthase Kinase-3 Beta (GSK3β)

GSK3β, a serine/threonine kinase, is integral in cancer due to its influence on a variety of cellular functions and signaling pathways. It plays a crucial role in regulating the cell cycle by phosphorylating proteins such as cyclin D1, which impacts cell proliferation [37]. When GSK3β is abnormally regulated, it can lead to excessive cell growth, commonly seen in cancer. In terms of apoptosis, GSK3β’s function varies; it can both promote and inhibit apoptosis depending on the cellular environment and its interactions with other molecular signals [38]. The PI3K/AKT pathway, which is often hyperactive in cancer, negatively regulates GSK3β [39]. AKT-mediated phosphorylation inhibits GSK3β, leading to enhanced cell survival and growth.

In the Wnt signaling pathway, GSK3β regulates β-catenin, a key component. When Wnt signaling is absent, active GSK3β, along with adenomatous polyposis coli (APC) and β-catenin, forms a complex with the Axin protein, leading to β-catenin’s phosphorylation and degradation. Conversely, active Wnt signaling inhibits GSK 3β, preventing β-catenin phosphorylation [40,41,42]. β-catenin also interacts with T-cell factor (TCF)/lymphocyte enhancer factor (LEF), promoting the transcription of genes associated with oncogenesis and cell migration, including c-Myc, cyclin D1, VEGF, and MMP-7 [43,44,45]. Recent studies highlight the significant role of GSK3β in maintaining the survival of normal cells, which is achieved through the mechanisms of the NF-κB pathway [46,47]. This enzyme, GSK3β, is not only crucial for normal cellular functions but also plays a supportive role in the survival of pancreatic cancer cells. The Mdm2 oncoprotein, which controls the p53 tumor suppressor protein, requires phosphorylation for p53 degradation. GSK3β is responsible for phosphorylating Mdm2, affecting p53’s stability [47]. Inhibiting GSK3β can prevent p53 degradation without impacting the Mdm2-p53 interaction or their cellular localization. Ionizing radiation, which is known to increase p53 levels, also causes GSK3β phosphorylation at serine 9, influencing p53 accumulation. A GSK3β mutant with an altered serine 9 reduces p53 and its target p21WAF-1’s activation, indicating that GSK3β 3 inhibition contributes to p53 stabilization by reducing Mdm2 phosphorylation following radiation exposure. GSK3β also interacts with other pathways, like Hedgehog and Notch, contributing to the development and progression of cancer [48]. Its role extends to influencing cancer metastasis, particularly affecting cell migration and invasion, which are essential for the spread of cancer cells. Given its central role in these processes, GSK3β is considered a potential therapeutic target. The development of inhibitors to suppress GSK3β is a focus in the quest to curb tumor growth and encourage apoptosis in cancer cells. However, targeting GSK3β is complex due to its diverse roles in different cancer types and its involvement in multiple signaling pathways. Selective inhibition that does not interfere with its normal physiological functions is crucial for effective cancer therapy.

In a detailed study conducted by Yasser Bustanji and his team, the focus was on elucidating the specific manner in which curcumin interacts with the binding pocket of GSK3β [49]. Their research utilized advanced molecular docking techniques to visualize and analyze the interaction between curcumin and GSK3β at a molecular level. The researchers found that the interaction of curcumin with GSK3β is characterized by a series of intricate binding interactions. Key among these is the involvement of curcumin’s conjugated enol–ketone system. The particular molecular structure of curcumin forms hydrogen bonds with the amino acid residue Val135, found in the binding pocket of GSK3β. Specifically, the enolic hydroxyl group of curcumin aligns itself to interact with the amidic carbonyl part of Val135. This interaction is significant as it suggests a strong and specific binding affinity between curcumin and GSK3β. Furthermore, the research highlights an interaction between the enolic oxygen atom of curcumin and the amino acid Ile62 within GSK3β. This interaction is mediated by two water molecules, indicating a more complex and indirect bonding mechanism. Such water-mediated interactions are often crucial in stabilizing the binding of small molecules within the active sites of larger enzymes. Figure 2 displays the roles of GSK3β and how they are suppressed by curcumin.

### 3.4. Dual-Specificity Tyrosine-Regulated Kinase 2 (DYRK2)

DYRK2 has a critical influence on the development and advancement of cancer [50,51]. It plays a key role in cell cycle management by phosphorylating essential proteins that facilitate the progress of cells through the cell cycle, thereby impacting cell division and proliferation [52]. DYRK2’s role in apoptosis, the programmed cell death mechanism vital for cancer cell elimination, is also noteworthy [53]. Its functioning can either enhance or suppress apoptosis, varying with the cellular environment and cancer type.

DYRK2 has been recognized as a crucial kinase, responsible for controlling the function of the 26S proteasome, a key entity in the degradation of proteins within eukaryotic cells. The reduction in DYRK2 levels, achieved via techniques such as si/shRNA or CRISPR/Cas9 knockout, leads to a marked decrease in proteasome efficiency. This decrease triggers proteotoxic stress, contributing to cell death, notably in breast cancer cells [54]. A vital aspect of DYRK2’s function involves the phosphorylation of the Rpt3 component in the proteasome’s 19S regulatory unit. This phosphorylation activity varies, intensifying during the G2/M phase of cell division and diminishing under certain conditions, such as serum deprivation [54]. The influence of DYRK2 extends to other cellular mechanisms, particularly in its phosphorylation of HSF1, a key transcription factor that plays a significant role in managing proteotoxic stress in cancerous cells. This phosphorylation by DYRK2 boosts HSF1’s stability and function within the nucleus, especially in TNBC cells, underscoring the kinase’s importance in maintaining cellular balance [55]. The activation of HSF1 through phosphorylation leads to the increased transcription of chaperone proteins, which aids in the proper folding of misfolded or unfolded proteins. Consequently, targeting DYRK2 inhibition, as seen with the effectiveness of the inhibitor LDN192960 in myeloma cells, emerges as a promising strategy in cancer therapy [56].

Furthermore, DYRK2 plays a crucial role in the cellular apoptosis pathway by participating in the ubiquitination process of the modulator of apoptosis protein 1 (MOAP-1). MOAP-1 is instrumental in activating Bax, a pro-apoptotic protein. The activation of Bax by MOAP-1 is a key step in the induction of apoptosis, a process of programmed cell death that is critical for maintaining cellular homeostasis and eliminating damaged or cancerous cells [57].

Sourav Banerjee and colleagues’ study demonstrates how curcumin selectively inhibits DYRK2 [58]. The crystallization of DYRK2 in the presence of curcumin, analyzed at 2.5 Å resolution, reveals that curcumin occupies the ATP-binding pocket of DYRK2. One of curcumin’s 4-hydroxy-3-methoxyphenyl groups forms hydrogen bonds with key amino acids of DYRK2, anchoring it within the ATP-binding pocket. These include Lys251 (forming an ion pair with Lys), Glu266 (forming an ion pair with Glu), and Asp368 (associated with the DFG Asp motif). Amino acids like Ile228, Ala249, Ile285, Phe301, Leu303, Leu355, and Ile367 engage in hydrophobic interactions with curcumin. The co-crystal structure of DYRK2 and curcumin elucidates that curcumin’s inhibitory effect on DYRK2 is due to its direct binding to the kinase’s ATP-binding pocket. Figure 2 showcases the functions of DYRK2 and its inhibition by curcumin.

**Figure 2 ijms-25-02911-f002:**
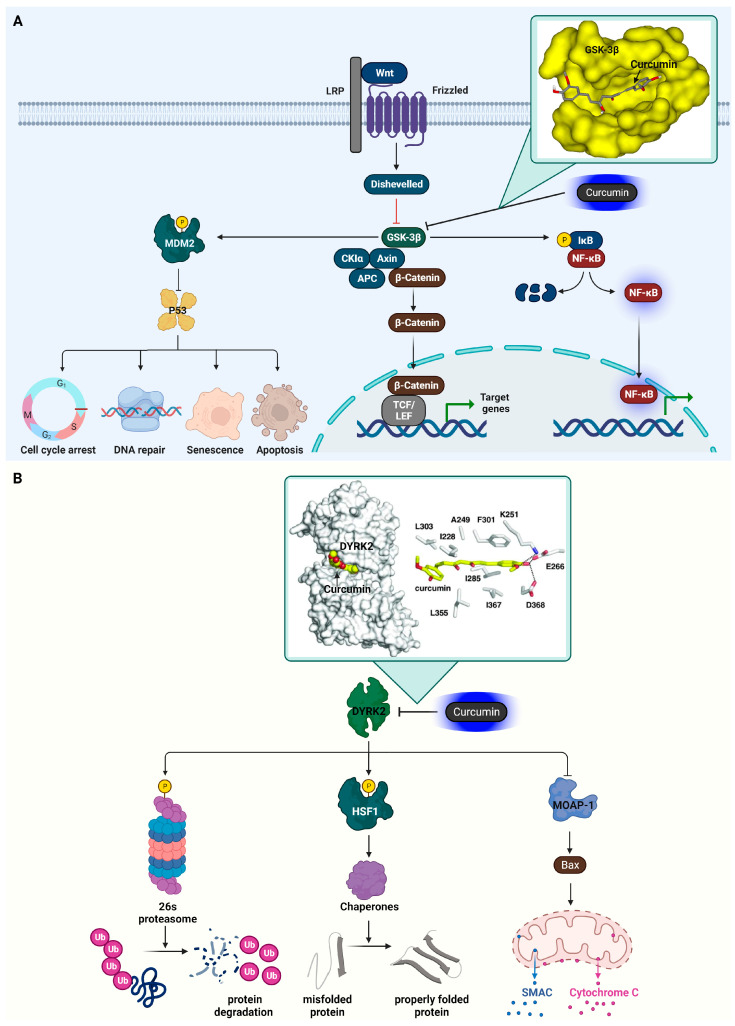
Curcumin’s inhibition of GSK3β and DYRK2 in cancer signaling pathways. (**A**) At the cell membrane, Wnt proteins attach to Frizzled receptors and LRP co-receptors, which activates the Dishevelled receptors. This activation results in the inhibition of GSK3β, thereby causing the stabilization and accumulation of β-catenin within the cytoplasm. Subsequently, β-catenin enters the nucleus and interacts with TCF/LEF transcription factors, influencing the expression of specific genes. Activation of the NF-κB pathway by GSK3β is mediated via the degradation of IκB. GSK3β contributes to the inhibition of p53-mediated cellular processes by promoting the phosphorylation of Mdm2. The inset details the molecular binding of curcumin to GSK3β, emphasizing how curcumin inhibits the enzyme by forming hydrogen bonds with the amino acid residue Val135. The docking illustration is referenced from [49]. (**B**) DYRK2’s pivotal role in enhancing the 26S proteasome’s function through phosphorylation, aiding in the correct refolding of misfolded proteins by triggering chaperone activity via HSF1 phosphorylation, is highlighted. Additionally, this enzyme plays a key role in cell death regulation by inhibiting Bax, a pro-apoptotic protein, through the suppression of MOAP-1. The inset focuses on the interaction between DYRK2 and curcumin, demonstrating how curcumin impedes the enzyme’s activity. This is specifically shown by the formation of hydrogen bonds between curcumin and DYRK2’s amino acid residues Lys251, Glu266, and Asp368, as illustrated in the docking image referenced from source [58]. The cartoon in Figure 2 was created with BioRender.com (https://app.biorender.com, accessed on 10 January 2024).

### 3.5. Epidermal Growth Factor Receptor (EGFR)

EGFR is a pivotal protein receptor in cancer, engaged in signaling pathways that oversee cell growth, division, and survival. Its activation and subsequent signaling play a key role in cancer biology. The activation of EGFR is triggered by the binding of ligands like epidermal growth factor (EGF) or transforming growth factor-alpha (TGF-alpha), leading to a series of signaling events that promote cellular proliferation, an essential aspect of normal cell growth and repair [59,60]. In various cancers, the abnormal expression, mutation, or misregulation of EGFR is common, resulting in uncontrolled cellular proliferation [61]. Such aberrant EGFR activity is characteristic of numerous cancer types, including lung, breast, and colorectal cancers.

Following its activation, EGFR engages in dimerization and autophosphorylation, initiating downstream signaling pathways such as the MAPK/ERK and PI3K/AKT pathways [62,63]. EGFR activation also leads to the stimulation of TGF-beta activated kinase 1 (TAK1) and mitogen-activated protein kinase kinase 3/6 (MKK 3/6), which are essential in cellular signaling [64]. Following this, there is a subsequent activation of P38, a key member of the MAP kinase family. The P38 pathway plays a significant role in regulating a variety of cellular processes, such as inflammation, cell growth, and apoptosis. Dysfunctional EGFR activation can drive tumor development, spread, resistance to apoptosis, and angiogenesis, fueling the tumor’s growth and metastasis. EGFR’s pivotal role in cancer progression makes it a primary target for therapeutic intervention. Treatments targeting EGFR, including tyrosine kinase inhibitors (TKIs) like gefitinib and erlotinib, as well as monoclonal antibodies such as cetuximab, are employed to combat cancers exhibiting EGFR anomalies [65]. A major challenge in targeting EGFR is the development of resistance to these inhibitors, often due to additional mutations or the activation of compensatory signaling pathways. Understanding EGFR’s complex signaling dynamics is vital for creating effective cancer treatments.

In the investigative study led by Amena Ali and colleagues, a detailed structural examination of the EGFR, particularly of the variant represented by PDB ID: 3W2R, was carried out [66]. This study focused on the interaction between EGFR and various curcumin analogs, which are known for their substantial molecular structures and weight. The research revealed that these curcumin analogs are suitably sized to fit into the active site of the EGFR molecule. Their interaction within this active site is characterized by a range of molecular interactions. These include hydrophobic contacts, which are critical for the stability of the compound within the enzyme’s active site; hydrogen bonding, which contributes to the specificity and strength of the binding; and π–π stacking interactions, a type of non-covalent interaction between the aromatic rings of the compounds and amino acids in the active site. One particular curcumin analog, labeled as 3a in the study, was noted for its unique molecular structure that incorporates a pyrimidine thione group. This specific structural feature of 3a allows it to engage in exclusive π–π stacking interactions with certain amino acids in the EGFR’s active site, namely Asp855, Asp800, and Leu718. These interactions are pivotal in determining the binding affinity and specificity of the analog towards EGFR. Another analog examined in the study, known as 3b, is distinguished by the addition of a pyrazoles group to the curcumin base structure. This modification enables 3b to form two hydrogen bonds with the amino acids Leu718 and Asp800 within the EGFR active site. In addition to these hydrogen bonds, 3b also establishes two π–π stacking interactions with the same amino acids. These interactions of 3b with the EGFR are indicative of its potential as a therapeutic agent, influencing the activity of the receptor in significant ways. Figure 3 illustrates the roles of EGFR and how it is inhibited by the curcumin analog, 3a.

### 3.6. AXL Receptor Tyrosine Kinase

The AXL receptor tyrosine kinase is linked to unfavorable outcomes in many cancers and the development of resistance to treatments [67,68]. It belongs to the TYRO3-AXL-MERTK kinase family. When it interacts with its specific ligand, GAS6, the AXL receptor influences a range of cell signaling pathways and the interaction among different elements of the tumor environment, such as cancer cells, endothelial cells, and cells of the immune system.

AXL receptor’s activation involves GAS6, a protein with a GLA domain, EGF-like repeats, and G domains, crucial for its interaction with AXL receptor [69,70]. The vitamin K-dependent γ-carboxylation of the GLA domains is essential for full TAM activation [71,72]. GAS6–AXL receptor complexes, forming a 2:2 stoichiometry, trigger AXL receptor’s dimerization and activation, leading to various intracellular signaling pathways like PI3K-AKT, and RAS-MEK-ERK [73,74,75]. Upon activation, the AXL receptor also activates JAK kinases, which phosphorylate specific tyrosine residues on AXL’s intracellular domain; these phosphorylated sites then serve as docking points for the STAT proteins that are phosphorylated by JAK, leading them to dimerize, enter the nucleus, and function as transcription factors for genes involved in cell growth, differentiation, and immune responses. Concurrently, this activation induces SOCS 1 and 3 proteins, creating a negative feedback loop that regulates cytokine signaling by inhibiting the JAK/STAT pathway through their binding to JAKs or the receptors, thus blocking further signal transduction [76]. Despite this canonical pathway, cancer cells often bypass GAS6 dependence [77]. AXL receptor overexpression and interaction with non-TAM RTKs like MET and EGFR diversify signaling in cancer cells [78]. The tumor suppressor OPCML can inactivate AXL receptor, hindering its interaction with other RTKs [79], underlining the need for further exploration of the AXL receptor’s complex, cancer-specific regulatory roles. Moreover, the AXL receptor plays a role in the epithelial-to-mesenchymal transition (EMT) process, a crucial phase in cancer metastasis [80,81]. EMT endows cancer cells with the ability to migrate and invade, enabling them to spread to distant body parts. The involvement of the AXL receptor in EMT highlights its significance in the metastatic mechanism of cancer. Given its crucial role in these cancer-related processes, the AXL receptor is being considered as a promising target for cancer therapy. Strategies to inhibit AXL receptor activity are being researched as a means to curb tumor growth, counteract drug resistance, and halt cancer metastasis. The development of inhibitors targeting the AXL receptor is a dynamic and important area in cancer research, aiming to enhance the efficacy of cancer treatments.

In their research, Fatima Ghrifi and colleagues utilized a 2D quantitative structure–activity relationship (2D-QSAR) methodology to examine 400 synthetic analogs of curcumin targeting the AXL receptor [82]. The study primarily focused on molecular docking scores, which ranged between −9.0 and −3.4 kcal/mol, indicating the molecular affinity of these compounds. Compounds demonstrating lower energy states and enhanced predicted activities were selected for an in-depth analysis of their interaction with the AXL receptor’s active site. Three specific compounds, identified as CID10765707, CID11257493, and CID21159180, were notable for their significantly low binding energies, suggesting strong activity predictions. These compounds underwent further evaluation to determine their interaction mechanisms with AXL. Key amino acids within a 4 Å radius of the AXL receptor active site, including Leu600, Pro621, Phe622, Met623, Lys567, Asp690, Phe691, and Gly692, were scrutinized in this context. The study found that these compounds engaged primarily in hydrogen and hydrophobic bonding with the site’s residues. Notably, Met 623 was observed to form hydrogen bonds with the hydroxyl groups of the phenyl rings in these compounds. The research indicated that these synthetic curcumin analogs exhibit a high affinity toward the AXL receptor, comparable to curcumin and its natural derivatives. Of these, CID21159180 (2-(Butylamino)-4-[(4-hydroxycyclohexyl) amino]-N-{[4-(1H-imidazol-1-yl)phenyl]methyl}pyrimidine-5-carboxamide) showed particularly promising activity levels. The study demonstrates the potential of these analogs as AXL receptor inhibitors, marking a significant step in the exploration of curcumin-based treatments in oncology. The signaling of the AXL receptor and its inhibition by CID21159180 are represented in Figure 3.

**Figure 3 ijms-25-02911-f003:**
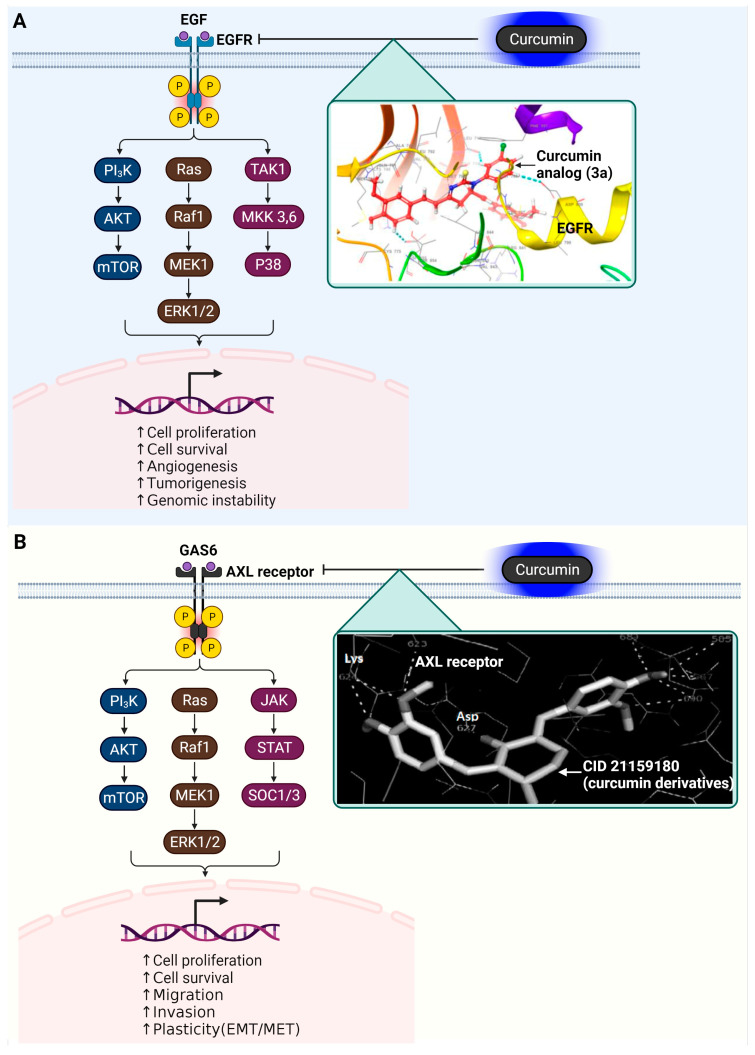
Impact of curcumin on EGFR and AXL receptor tyrosine kinases in oncogenic signaling pathways. (**A**) EGFR, following dimerization and autophosphorylation, triggers several downstream pathways, including MAPK/ERK and PI3K/AKT. The activation of EGFR also stimulates TAK1 MKK 3/6, leading to the activation of P38, a vital part of the MAP kinase family. An inset illustrates the molecular interaction between curcumin and EGFR, highlighting curcumin’s inhibition of the enzyme through π–π stacking interactions with residues Asp855, Asp800, and Leu718, as indicated in the docking studies referenced in [66]. (**B**) The binding of GAS6 to the AXL receptor forms complexes that activate intracellular signaling pathways such as PI3K-AKT, RAS-MEK-ERK, and JAK-STAT, SOCS1/3. An inset displays the molecular interaction between CID 21159180 (a derivative of curcumin) and the AXL receptor, indicating how curcumin inhibits the enzyme by forming hydrogen bonds with Met623, Glu585, and Asp627, as detailed in the docking illustration from [82]. (**B**) DYRK2’s pivotal role is highlighted; it enhances the 26S proteasome’s function through phosphorylation, aiding in the correct refolding of misfolded proteins by triggering chaperone activity via HSF1 phosphorylation. Additionally, this enzyme plays a key role in cell death regulation by inhibiting Bax, a pro-apoptotic protein, through the suppression of MOAP-1. The detailed view in the inset focuses on the interaction between DYRK2 and curcumin, demonstrating how curcumin impedes the enzyme’s activity. This is specifically shown by the formation of hydrogen bonds between curcumin and DYRK2’s amino acid residues Lys251, Glu266, and Asp368, as illustrated in the docking image referenced from [58]. The cartoon in Figure 3 was created with BioRender.com (https://app.biorender.com, accessed on 10 January 2024).

### 3.7. Folate Receptor β (FR-β)

FR-β holds a crucial position in the progression of cancer, particularly within the tumor microenvironment [83]. As an important part of the folate receptor group, including FR-α, FR-γ, and FR-δ, FR-β is notably expressed at high levels in tumor-associated macrophages (TAMs), particularly the M2-polarized subtype [84]. These M2-polarized TAMs, a major segment of leukocytes in the tumor environment, play a role in promoting tumor invasion, growth, angiogenesis, metastasis, and immunosuppression. The marked expression of FR-β in M2-polarized TAMs highlights its potential as a target for cancer treatment. TAMs are dynamic, with the ability to either suppress tumor growth (M1 macrophages) or enhance it (M2 macrophages). The presence of FR-β in lung cancer has been linked to the prognosis of the disease. Cancer treatment strategies that focus on TAMs seek to decrease their number and modify their function, impacting cancer’s progression. Targeting FR-β offers a strategy for selectively delivering medication to these macrophages, which has been supported by studies demonstrating the success of folate–drug conjugates [85]. Experimental evidence has shown that targeting FR-β with specific treatments, such as recombinant immunotoxins, can substantially reduce TAMs and inhibit tumor growth [86]. Treatments like zoledronic acid encapsulated in folate-modified liposomes have been effective in inducing targeted cytotoxicity through folate receptors in a controlled environment. Despite this, targeted therapies specifically using FR-β for lung cancer TAMs remain in development.

Additionally, Folate binding to FR-β can activate signaling pathways, such as STAT3 activation via a GP130 co-receptor-mediated JAK-dependent process [87]. Folate can also bind to folate receptors that undergo endocytosis. Folate receptors have an additional role in transporting folate into cells through a specialized endocytosis process known as potocytosis, bypassing the clathrin-coated pits pathway [87]. Once internalized, folate receptors can move to the nucleus, functioning as transcription factors by attaching to cis-regulatory sequences. The tetrahydrofolate (THF), once freed from the endosome, is converted into formate, which is utilized for biosynthetic processes, including the formation of purines and thymidylate, which are essential for DNA synthesis and repair.

In the research conducted by Barbara Frigerio and colleagues, molecular docking studies were undertaken to examine the interaction of curcumin with FR-β [88]. The results indicate that curcumin establishes hydrogen bonds with specific amino acids at FR-β’s active site, including Asp97, Ser190, Arg152, and His151. These bonds are formed primarily through the phenol groups and the hydroxyl group in the curcumin’s linker chain. Additionally, curcumin appears to engage in π-stacking interactions with Tyr-101 and forms double π-stacking interactions with Trp187 within the active site. The findings from these docking studies highlight curcumin’s potential affinity for FR-β, as evidenced by a notable docking score of −63.30 arbitrary units. This score suggests a strong binding propensity of curcumin towards FR-β, indicating its potential as a significant molecule in targeting this receptor. The signal transduction of FR-β and its modulation by curcumin are illustrated in Figure 4.

### 3.8. Dihydrofolate Reductase (DHFR)

DHFR is a pivotal enzyme in cancer biology due to its role in DNA and RNA synthesis. It facilitates the conversion of dihydrofolate into THF, a necessary step for producing thymidylate, purines, and certain amino acids. These components are essential for the synthesis of DNA and RNA, playing a vital role in cell division and growth [89,90]. In cancer cells, which divide at an accelerated rate compared to normal cells, DHFR’s role becomes increasingly significant [91,92]. The enhanced requirement for DNA synthesis in these cells heightens their reliance on the folate cycle, of which DHFR is a key component. This dependency positions DHFR as a strategic target in cancer chemotherapy. Chemotherapy agents that inhibit DHFR, like methotrexate, are critical as they interrupt the DNA and RNA synthesis in cancer cells, leading to their death. However, these agents also impact normal cells that divide quickly, resulting in common chemotherapy side effects such as hair loss and gastrointestinal issues. Targeting DHFR in chemotherapy aims to reduce or halt the proliferation of cancer cells, an essential strategy in managing various cancer types. The efficacy of DHFR inhibitors in treating cancer underscores the enzyme’s significance in the progression of the disease and its value as a target for therapeutic intervention [93].

In their research, Yahya Hobani and colleagues conducted a molecular docking study of curcumin with DHFR, a key enzyme in cell proliferation [94]. The docking results showed that curcumin had a total free energy of binding of −9.02 kcal/mol and an estimated inhibition constant (KI) of 243 nM, indicating a strong affinity for DHFR. The study’s graphical depictions of the curcumin–DHFR complex highlighted that curcumin adopts a bent conformation within the DHFR active site. This positioning allows for extensive van der Waals interactions with the enzyme’s surrounding residues. Notably, a phenyl ring in curcumin engages in pi–pi stacking interactions with Phe-34 within the active site, reflecting the hydrophobic interaction characteristics of folate and methotrexate (MTX), a known DHFR inhibitor, binding to DHFR. One aromatic ring of Curcumin, specifically ring B, engages in pi–pi stacking with the aromatic ring of Phe34 within DHFR’s active site. The hydroxyl group of Curcumin’s side chain A forms hydrogen bonds with both the NH and CO groups of Ala9, and with the two oxygen atoms in the carboxyl group of Glu30 in DHFR. Additionally, the phenolic hydroxyl group on Curcumin’s A ring establishes hydrogen bonding with the CO group of Glu30 and with both the CO and NH groups of Phe31 in DHFR. Curcumin also creates hydrogen bonds through its side chain B hydroxyl group with the CO group of Val115 and the phenolic hydroxyl group of Tyr121 in DHFR. This effective binding suggests a strong and significant interaction between curcumin and DHFR, underlying curcumin’s potential as an inhibitor of this enzyme. The metabolic functions of DHFR and their suppression by curcumin are presented in Figure 4.

**Figure 4 ijms-25-02911-f004:**
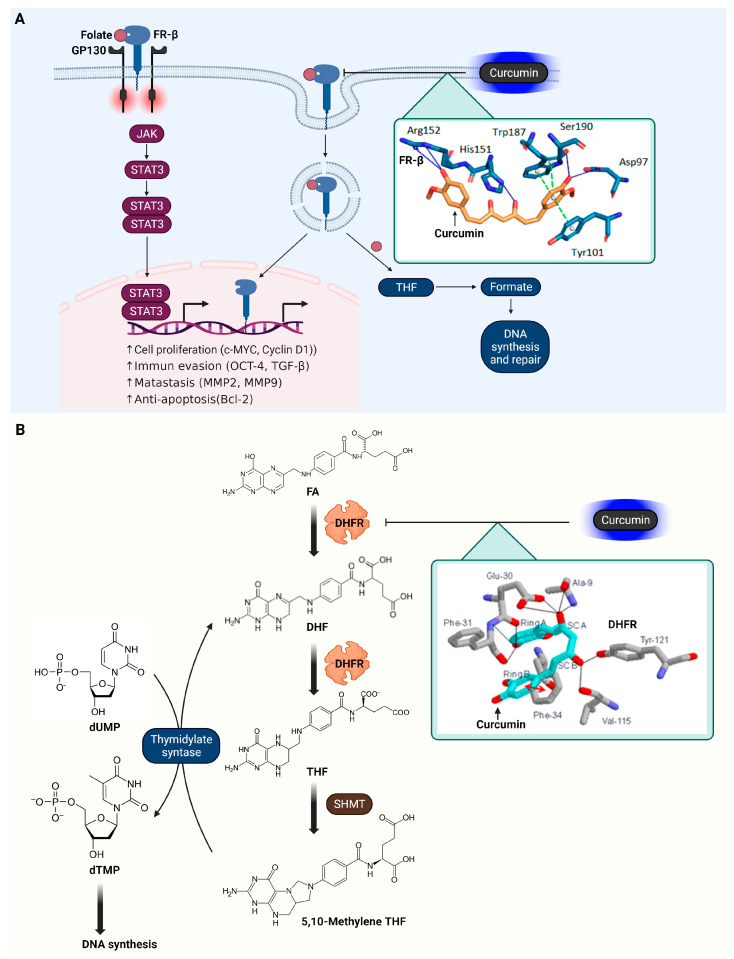
Effects of curcumin on FR-β and DHFR in cancer-related signaling pathways. (**A**) The interaction of folate with FR-β activates pathways, including STAT3, through a GP130 co-receptor mediated JAK-dependent mechanism. Additionally, folate receptors, when bound to folate, can undergo endocytosis and migrate to the nucleus, acting as transcription factors. Within the endosome, THF is converted to formate, used in biosynthesis for creating purines and thymidylate, which are crucial for DNA synthesis and repair. An inset details the interaction between curcumin and FR-β, where curcumin inhibits the enzyme via π–π stacking with Tyr-101 and Trp187 and forms hydrogen bonds with Asp97, Ser190, Arg152, and His151, as shown in the referenced docking studies [88]. (**B**) DHFR plays a role in converting dihydrofolate to THF, which is essential for synthesizing thymidylate, purines, and certain amino acids. The inset shows the interaction of curcumin with DHFR, illustrating how curcumin impedes the enzyme by forming hydrogen bonds with Ala9, Glu30, Phe31, Val115, and Tyr121, as documented in the docking study [94]. The cartoon in Figure 4 was created with BioRender.com (https://app.biorender.com, accessed on 10 January 2024).

### 3.9. DNA Topoisomerase I (Topo I) and II (Topo II)

Topo I and Topo II are pivotal in cancer management due to their role in DNA topology regulation during key cellular activities. Topo I primarily functions to relax supercoiled DNA by cleaving a single DNA strand, enabling its unwinding, followed by reconnection. This mechanism is crucial in alleviating the torsional stress encountered during DNA replication and transcription. By relaxing DNA supercoils, Topo I facilitates the smooth progression of the transcription and replication machinery, which is essential for RNA synthesis and DNA replication [95,96]. Topo I also serves as a target for specific cancer drugs like topotecan and irinotecan, which interfere with the Topo I cycle’s re-ligation phase, causing DNA damage and potentially leading to cell death in fast-proliferating cancer cells [97].

Topo II, on the other hand, addresses DNA tangles and supercoils during replication and transcription by cutting double-strand DNA [98,99]. It is vital for segregating replicated DNA helices and plays a key role in chromosome condensation and segregation during mitosis, ensuring accurate genetic material distribution. Similar to Topo I, Topo II is targeted by several anticancer drugs, including etoposide and doxorubicin [98,100]. These medications stabilize the double-strand breaks made by Topo II in DNA, causing cell cycle arrest and apoptosis in cancer cells.

Both DNA topoisomerases are essential to DNA’s structural integrity during critical processes like replication and transcription. Their significant functions in DNA dynamics render them crucial targets in cancer treatment, as inhibiting them can cause DNA damage and cell death in cancer cells.

Anil Kumar and colleagues’ molecular docking study highlights the interaction of curcumin natural derivatives with Topo I and II [101]. The study reveals that these derivatives bind at the DNA cleavage site, parallel to the DNA base-pairing axis. The binding efficacy of cyclocurcumin (C_21_H_20_O_6_) to Topoisomerase I, with PDB ID: 1K4T, is quantified by a Gibbs free energy change (ΔG) of −10.33 kcal/mol, signifying potent molecular interaction. This association is distinguished by polar contacts involving the amino acid residues Asp479, Ser480, and Gln778, as well as interactions with the nucleotide bases T10 and A-113, both highlighted in magenta, within the critical binding domain of the enzyme. Similarly, the interaction of cyclocurcumin with Topoisomerase II, represented by PDB ID: 3QX3, yields a Gibbs free energy change (ΔG) of −10.33 kcal/mol. This denotes a comparably strong binding affinity, with cyclocurcumin forming polar interactions at the same amino acid residues—Asp479, Ser480, and Gln778—and additionally with the nucleotide bases T9 and G10, which are also marked in magenta within the enzyme’s active site. The unwinding of DNA supercoils by Topo I and Topo II and their inhibition by cyclocurcumin are illustrated in Figure 5.

### 3.10. Nuclear Factor Kappa-Light-Chain-Enhancer of Activated B Cells (NF-κB)

In cancer, NF-κB significantly influences cell proliferation and survival by regulating the expression of specific target proteins. When activated, NF-κB transcribes genes that lead to the production of proteins like cyclin D1 [102], which is crucial for cell cycle progression, and Bcl-2 and Bcl-xL, known for their roles in inhibiting apoptotic pathways [103]. These proteins collectively contribute to the enhanced survival and growth of cancer cells.

NF-κB’s role in inflammation and cancer is closely linked, as it promotes a chronic inflammatory environment that can lead to oncogenesis. This involves the activation of genes that encode pro-inflammatory cytokines like tumor necrosis factor-alpha (TNF-α) and interleukin-6 (IL-6), which not only sustain inflammatory responses but also support tumor growth and survival [104].

The activation of NF-κB in cancer is triggered by various stimuli, including inflammatory cytokines, growth factors, and stress signals. This activation initiates the transcription of genes involved in critical processes such as cell proliferation, survival, angiogenesis, metastasis, and inflammation. For instance, in angiogenesis, NF-κB upregulates vascular endothelial growth factor (VEGF), promoting the formation of new blood vessels necessary for tumor growth and metastasis [105]. Additionally, NF-κB’s role in apoptosis and drug resistance is marked by its regulation of anti-apoptotic genes. By inhibiting pro-apoptotic proteins like caspases and promoting anti-apoptotic proteins such as the FLICE-like inhibitory protein (c-FLIP) and inhibitors of apoptosis proteins (IAPs), NF-κB renders cancer cells more resistant to cell death and, consequently, to various chemotherapeutic agents [106]. Overall, NF-κB’s impact on cancer involves a complex network of gene regulation, affecting a range of proteins that are central to cancer cell survival, proliferation, and resistance to treatment. Understanding these specific interactions and the proteins that are involved is crucial for developing targeted therapies against cancer.

In their research, Mohamed EM Saeed and colleagues conducted a comprehensive molecular docking study involving 50 curcumin derivatives, sourced from the PubChem database [107]. These compounds were analyzed for their binding affinity with NF-κB. The results of the docking study indicated a range of binding energies for NF-κB with these curcumin compounds, spanning from −12.97 (±0.47) to −6.24 (±0.06) kcal/mol. This range suggests varying degrees of interaction strength between the different curcumin derivatives and NF-κB.

Additionally, research led by Sompot Jantarawong explored the detailed interactions between curcumin and the p50 subunit of NF-κB [108]. Their findings revealed a variety of interaction types, including both classical and non-classical hydrogen bonds, such as carbon and π-donor interactions. The study also identified significant electrostatic interactions involving π-charge (including π-cation and π-anion) interactions, as well as alkyl, mixed π/alkyl hydrophobic interactions, and π-sulfur interactions. These interactions contribute to the stability and specificity of curcumin’s binding to NF-κB. Notably, the study pinpointed specific amino acid residues in the p50 subunit of NF-κB that interact with curcumin. These residues include Thr 256, Ala 257, Pro 324, Pro 344, Phe 345, and Leu 346. The interaction with these residues suggests potential sites where curcumin could influence the activity of NF-κB, providing insights into how curcumin derivatives might modulate the function of this crucial transcription factor in the context of cancer and other diseases where NF-κB plays a key role. Table 1 displays the proteins targeted by curcumin in cancer. The activation of NF-kB signaling and the inhibition of NF-kB’s activity are portrayed in Figure 5. The 10 target proteins of curcumin in cancer, as previously discussed, are listed in Table 1.

**Figure 5 ijms-25-02911-f005:**
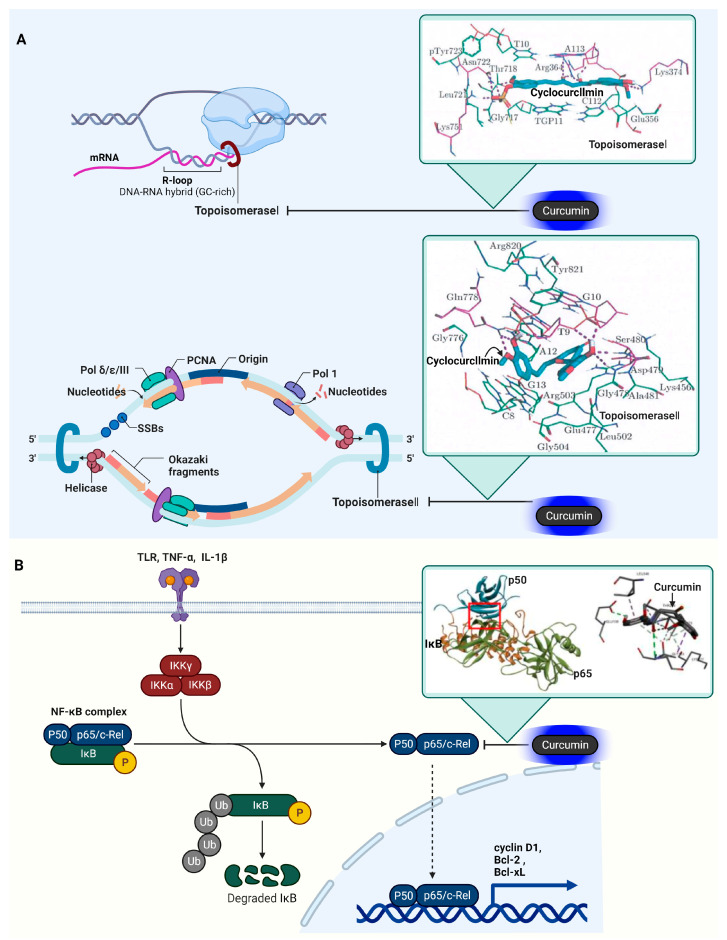
Influence of curcumin on Topo I, II, and NF-κB in oncogenic signaling pathways. (**A**) Topo I is involved in relaxing supercoiled DNA through single-strand cleavage, while Topo II resolves DNA entanglements and supercoils during replication and transcription by cleaving double-stranded DNA. An inset describes the molecular interaction between curcumin and both Topo I and Topo II, focusing on curcumin’s interaction with Arg364, Asn722, and base A113 in the Topo I–DNA complex, and Asp479, Gln778, and base T9 in the Topo II–DNA complex, as indicated in the docking studies [101]. Red square indicates RNA primer. (**B**) The phosphorylation process involving IKK, IκB, and NF-κB is a critical pathway in cellular signaling. Initially, IKK is activated, often in response to various stimuli like cytokines or stress factors. Once activated, IKK phosphorylates the inhibitor of κB (IκB), a protein that is bound to NF-κB in its inactive state in the cytoplasm. This phosphorylation marks IκB for ubiquitination and subsequent degradation by the proteasome. With the degradation of IκB, NF-κB is released from its inhibited state. Free NF-κB, which is now active, can then translocate into the nucleus, where it binds to specific DNA sequences to regulate the transcription of genes involved in immune and inflammatory responses, cell growth, and apoptosis. Blue arrows indicate gene expression. The inset presents the interaction between curcumin and NF-κB, highlighting how curcumin blocks the enzyme’s function by forming hydrogen bonds with Thr 256, Ala 257, Pro 324, Pro 344, Phe 345, and Leu 346, as elucidated in the docking study [108]. The cartoon in Figure 5 was created with BioRender.com (https://app.biorender.com, accessed on 10 January 2024).

## 4. Curcumin Target Proteins in Inflammation

The connection between cancer and inflammation is both intricate and consequential. Chronic inflammation is increasingly acknowledged as a pivotal element in cancer’s onset and progression. The process begins with chronic inflammation, potentially triggering cancer by causing DNA damage in cells due to inflammatory cells and cytokines [109,110]. This damage can start the carcinogenesis process if left unrepaired. Furthermore, inflammation fosters an environment suitable for tumor growth by releasing specific cytokines, such as IL-6 and TNF-α, and growth factors like vascular endothelial growth factor (VEGF), which enhance cell proliferation and aid in tumor nourishment through angiogenesis [111].

Moreover, inflammation can facilitate cancer metastasis by releasing specific cytokines such as IL-8 and TNF-α, which increase the invasiveness of tumor cells and prepare distant tissues for their arrival, thus promoting metastatic spread [112]. While the immune system typically targets and destroys emerging cancer cells, chronic inflammation can impair this mechanism through various mechanisms, such as by inducing the production of immunosuppressive cytokines like IL-10 and transforming growth factor-beta (TGF-β) [113,114]. These cytokines can reduce the activity of cytotoxic T cells and natural killer cells, crucial components of the immune system’s anti-cancer response. Additionally, chronic inflammation can promote the recruitment and activation of myeloid-derived suppressor cells (MDSCs) and regulatory T cells (Tregs), which further suppress the immune response, thereby allowing cancer cells to evade immune detection and destruction [115,116]. Cancer cells often activate inflammatory signaling pathways, such as NF-κB, STAT3, and COX-2, which can lead to increased tumor growth and resistance to cell death [117].

Cancer treatments themselves can induce inflammation, potentially causing side effects and even aiding in tumor recurrence [118]. Understanding the dynamic between inflammation and cancer is vital for developing effective prevention and treatment strategies, with anti-inflammatory drugs currently being explored for their cancer-preventive potential. In this context, research on curcumin, which is known for its anti-inflammatory properties, is crucial, particularly in molecular docking studies. These studies can reveal how curcumin interacts with specific inflammatory molecules, offering insights into its anti-inflammatory mechanisms. Identifying new molecular targets for curcumin in inflammatory pathways could lead to more effective therapies. Molecular docking is vital in designing curcumin-based drugs to improve their efficacy and bioavailability, overcoming limitations like low bioavailability. Curcumin’s interaction with inflammation-related molecules like COX-2, phosphodiesterase 4, and C-reactive protein opens avenues for new treatments for various diseases driven by chronic inflammation. The continuing research in this area holds significant promise for healthcare advancements in treating inflammatory diseases.

### 4.1. Cyclooxygenase-2 (COX-2)

COX-2, a vital enzyme in the inflammation pathway, plays a crucial role in synthesizing prostaglandins, which are central to the inflammatory response [119]. This enzyme is distinct from COX-1, which is constantly present in most tissues for normal functioning. COX-2 is typically induced during inflammation, catalyzing the production of prostaglandins, particularly prostaglandin E2 (PGE2), from arachidonic acid, which are responsible for symptoms such as pain, swelling, and redness [120]. Its induction is triggered by inflammatory stimuli, including cytokines like IL-1 and TNF-α, growth factors, and endotoxins. COX-2’s pivotal role in inflammation makes it a prime target for NSAIDs, aiming to ease inflammation and pain by blocking COX-2 activity. Its involvement in various inflammatory diseases like rheumatoid arthritis and osteoarthritis has led to the development of selective inhibitors that target COX-2 specifically, reducing the gastrointestinal side effects typical of traditional NSAIDs [121].

Immune cells such as macrophages, T cells, and dendritic cells are at the forefront of inflammatory responses, which result in the release of reactive species, including ROS and RNI, and a range of cytokines like TNF-α, IL-6, IL-1, and IL-17A [122,123]. These reactive agents are crucial for causing genetic and epigenetic changes that may give rise to and support the proliferation of transformed cells, a critical early phase in the formation of tumors. Furthermore, the signaling pathways activated by these agents, NF-κB, STAT3, MAPK, and Akt, are central to the progression of cancer [124,125]. They orchestrate a variety of oncogenic processes, including cancer cell growth, survival, invasion, and angiogenesis, which are necessary for the tumor’s growth and metastasis. The inflammatory response thus plays a complex role in cancer: it can defend the body against pathogens but also, paradoxically, facilitate the development and propagation of cancer. This complexity presents a significant opportunity for developing targeted cancer therapies that can mitigate the unintended consequences of immune responses.

Subhash Padhye’s team conducted extensive molecular docking studies to examine how curcumin and its analogs interact with COX-2, a key enzyme in both inflammation and cancer [126]. Notably, curcumin demonstrated a single hydrogen bond with the amino acid residue Ala562 within the COX-2 enzyme. This specific interaction was significant in terms of the enzyme’s inhibition. The binding energy calculated for curcumin was −5.71 kcal/mol, indicating a stable interaction, although not the strongest. On the other hand, a more potent curcumin analog, curcumin difluorinated (CDF), exhibited a more complex and stronger interaction with the COX-2 enzyme. CDF was found to form four hydrogen bonds with different residues of COX-2. These residues were Glu346, Phe580, Asn101, and Gln350. Each of these interactions contributes to the stability and effectiveness of CDF in inhibiting COX-2. The binding energy of CDF was notably higher at −7.91 kcal/mol, reflecting a more potent interaction compared to curcumin. These findings indicate that while curcumin itself interacts with COX-2 and potentially inhibits its function, its analog CDF shows a more substantial binding affinity, suggesting a higher potential for therapeutic use in conditions where COX-2 inhibition is beneficial, such as in certain types of cancer and inflammatory diseases. This research underscores the importance of molecular docking studies in the development of more effective curcumin-based drugs, especially for targeting key proteins, like COX-2, in disease treatment. The induction of inflammatory responses through COX-2 and its interaction with curcumin are depicted in Figure 6.

### 4.2. C-Reactive Protein (CRP)

CRP plays a critical role in the body’s inflammatory response. As an acute-phase protein, its levels rise dramatically during inflammatory processes, acting as a biomarker for inflammation. CRP is produced by the liver in response to factors released by fat cells (adipocytes) and by the presence of certain pro-inflammatory cytokines, such as IL-6 and TNF-α. In inflammation, CRP’s primary function is to bind to phosphocholine expressed on the surface of dead or dying cells (and some types of bacteria), thereby activating the complement system, which is part of the body’s innate immunity. This activation leads to the opsonization of pathogens and dead cells, marking them for clearance by macrophages and neutrophils. Moreover, studies have shown that CRP elevates the production of cytokines such as IL-1 and TNF-α in human monocytes and alveolar macrophages [127]. CRP also engages with the FcγRII (CD32) receptor on endothelial cells, initiating a cascade of cellular reactions. This interaction with CRP stimulates the MAPK pathway, subsequently leading to the phosphorylation and activation of the transcription factor P65. Once activated, P65, a subunit of the NF–κB complex, moves into the nucleus and modulates the transcription of genes involved in the inflammatory response [128]. Inflammation, as a tumor-promoting factor, contributes to a microenvironment conducive to cancer progression. In the context of cancer, CRP’s role is more complex. Elevated CRP levels have been observed in various types of cancers, often correlating with tumor progression and poorer prognosis [129]. This elevation in CRP is thought to reflect the systemic inflammatory response to the tumor. CRP may indirectly influence tumor growth and metastasis by affecting the tumor microenvironment, modulating immune responses, and influencing the behavior of cancer cells. Inflammatory cytokines in the tumor microenvironment can stimulate CRP production, which, in turn, may impact tumor cell proliferation, angiogenesis, and metastasis. Thus, CRP serves not only as a biomarker for cancer but also as a potential link between chronic inflammation and cancer progression.

Neda Shakour and colleagues investigated the binding affinity of four curcumin variants (curcumin, cyclocurcumin, demethoxycurcumin, bisdemethoxycurcumin) with CRP [130]. The docking studies revealed that curcumin forms a hydrogen bond with the CRP’s Gln150 residue via its carbonyl group and another with Asp140 through its hydroxyl group, resulting in a pKi value of 13.12. Cyclocurcumin showed a pKi of 12.40, binding through a hydrogen bond from its hydroxyl group to the Lys28 residue. Demethoxycurcumin, with a pKi of 9.40, bonded via an oxygen atom from its methoxy group to the Ser68 residue. Bisdemethoxycurcumin displayed a pKi of 8.85, interacting through a hydrogen bond between its carbonyl group and the Gln150 residue. Additionally, an X-ray phosphorylcholine interaction with CRP, featuring a hydrogen bond to the Asn61 residue, had a pKi of 14.63. The docking data highlighted that curcumin and bisdemethoxycurcumin share a common hydrogen bond interaction at the Gln150 site. The role of CRP in inducing inflammation and its inhibition by curcumin are depicted in Figure 6.

**Figure 6 ijms-25-02911-f006:**
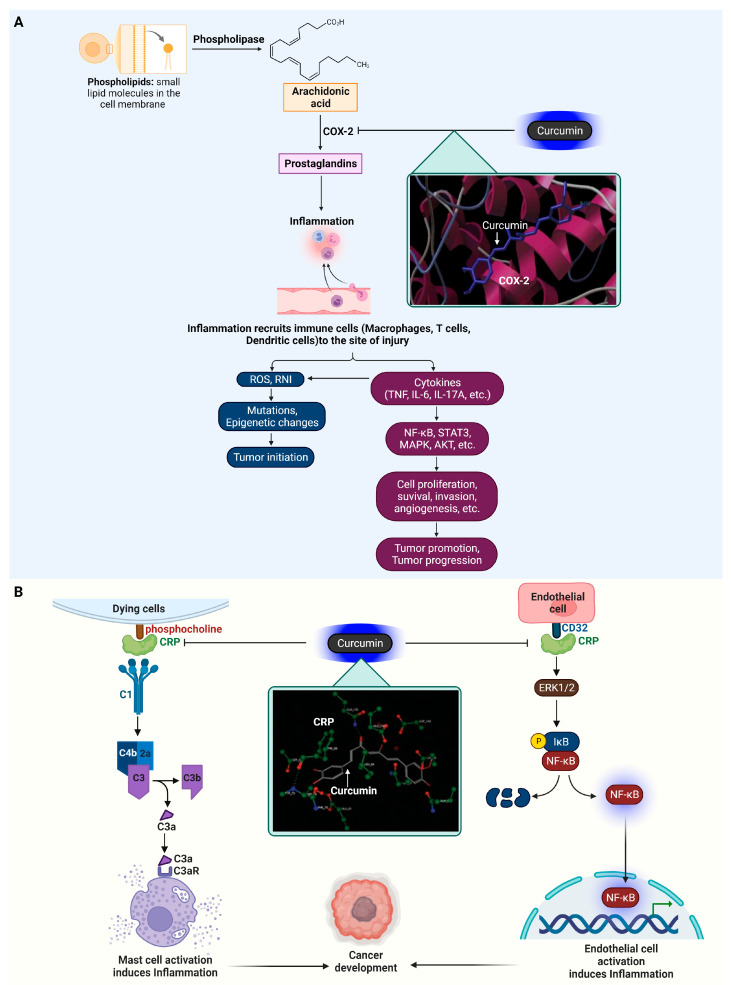
Curcumin’s influence on COX-2 and CRP in inflammatory processes. (**A**) Exposure to inflammatory stimuli such as cytokines (IL-1, TNF-α), growth factors, and endotoxins leads to the upregulation of COX-2, which then converts arachidonic acid into prostaglandins. These prostaglandins contribute to the release of reactive species including ROS and RNI, along with cytokines like TNF-α, IL-6, IL-1, and IL-17A. An inset highlights the interaction between curcumin and COX-2, specifically the formation of hydrogen bonds with Ala562, as revealed in the docking studies [126]. (**B**) CRP, binding to phosphocholine on the surfaces of dead or dying cells and certain bacteria, triggers the complement system. It also interacts with the FcγRII (CD32) receptor on endothelial cells, stimulating the MAPK pathway and leading to the activation of NF-κB. The inset demonstrates how curcumin interacts with CRP, notably impeding its function by forming hydrogen bonds with Gln150 and Lys28, as established in the docking study [130]. The cartoon in Figure 6 was created with BioRender.com (https://app.biorender.com, accessed on 10 January 2024).

### 4.3. Phosphodiesterase 4 (PDE4)

PDE4 significantly influences inflammation regulation by affecting cyclic adenosine monophosphate (cAMP) levels in cells. As an essential secondary messenger, cAMP is involved in numerous cell processes, notably in managing inflammation [131,132].

PDE4’s activity is modulated through its interactions with β-adrenoceptors, NMDARs, and 5-HT receptors [133]. These interactions lead to an increase in cAMP levels through the activation of adenylate cyclase, which subsequently activates protein kinase A (PKA) and cAMP-activated exchange proteins 1/2 (Epac1/2) [134]. PKA activation results in the phosphorylation of the cAMP-responsive element-binding protein (CREB) and activating transcription factor 1 (ATF-1), thereby boosting the production of anti-inflammatory cytokines. PKA’s action also extends to modifying the transcriptional activity of NF-κB by influencing its interaction with the CREB binding protein (CBP) or p300 [135]. Furthermore, PKA can interfere with the generation of pro-inflammatory cytokines controlled by B-cell lymphoma 6 protein (Bcl-6) and can impact immune cell proliferation [136]. The spatial distribution of cAMP in cells is crucial for the function of the Epac signalosome, which includes transcription factors and small GTPases like Rap1. This mechanism provides an alternative strategy for managing inflammation and cellular proliferation.

In cancer, the role of PDE4 extends to influencing cAMP signaling in tumor cells, which may affect cancer progression and the immune system’s ability to combat tumors [137]. This makes PDE4 a promising target in cancer treatment strategies. In essence, PDE4 is a vital enzyme in controlling inflammation. Targeting PDE4 to modulate the cAMP pathway presents a therapeutic approach for treating a range of chronic inflammatory conditions and potentially some cancers, aiming to lessen inflammation and curb tumor development.

Veronika Furlan and her team conducted research on the molecular interactions between curcumin and PDE4D (with the PDB ID: 3iad), employing the CANDOCK docking software (available at https://github.com/chopralab/candock_benchmark, accessed on 10 January 2024) for their analysis [88]. They discovered that curcumin forms several key interactions at the active site of PDE4D. The docking score for this interaction was found to be −62.24 kcal/mol, indicating a strong affinity between curcumin and PDE4D. Specifically, curcumin established hydrogen bonds with the amino acid residues His336, Asn375, Met439, and Asn602 in PDE4D. In addition to these hydrogen bonds, Met439 was also involved in forming a hydrophobic interaction with curcumin. These findings suggest that curcumin’s binding at the active site of PDE4D involves a combination of both hydrogen bonding and hydrophobic interactions, contributing to its potential inhibitory effect on the enzyme. The function of PDE4D in mediating inflammatory reactions and its suppression by curcumin are presented in Figure 7.

### 4.4. Myeloid Differentiation Protein 2 (MD-2)

In inflammation, MD-2 plays a crucial role in the body’s innate immune response [138,139]. MD-2 is a key component of the Toll-like receptor 4 (TLR4) complex, which is essential for recognizing and responding to lipopolysaccharides (LPS) from Gram-negative bacteria. Functionally, MD-2 binds to TLR4, and this interaction is necessary for the TLR4 complex to recognize LPS. Upon encountering LPS, MD-2 facilitates the binding of LPS to TLR4, triggering a series of downstream signaling events [140]. These events involve the recruitment of adaptor proteins like MyD88 and TRIF. The MyD88-dependent pathway leads to the activation of the NF-κB and the production of pro-inflammatory cytokines such as TNF-α, IL-6, and IL-1. These cytokines play a crucial role in initiating and propagating the inflammatory response. In parallel, the TRIF-dependent pathway leads to the activation of interferon regulatory factor 3 (IRF3) and the production of Type I interferons. This pathway plays a role in antiviral responses and further amplifies the inflammatory response.

Zhe Wang and their team utilized AutoDock to analyze the crystal structure of the human MD2–lipid IVa complex (PDB code 2E59) [141]. They investigated the binding affinity of curcumin to MD2 through surface plasmon resonance (SPR) experiments. These experiments demonstrated that curcumin binds to MD2 in a dose-dependent manner, showing a notably high affinity with a dissociation constant (KD) of 0.000379 M. Their study further revealed that curcumin can fit into the large hydrophobic binding pocket of MD2, extensively covering the area where LPS usually bind. Additionally, curcumin was found to establish multiple hydrogen bonds with specific amino acid residues of MD2, namely Arg90, Glu92, and Tyr102, contributing to its strong binding affinity. The involvement of MD-2 in triggering inflammatory responses and its suppression by curcumin is illustrated in Figure 7. The four target proteins of curcumin in inflammation, as previously reviewed, are presented in Table 2.

**Figure 7 ijms-25-02911-f007:**
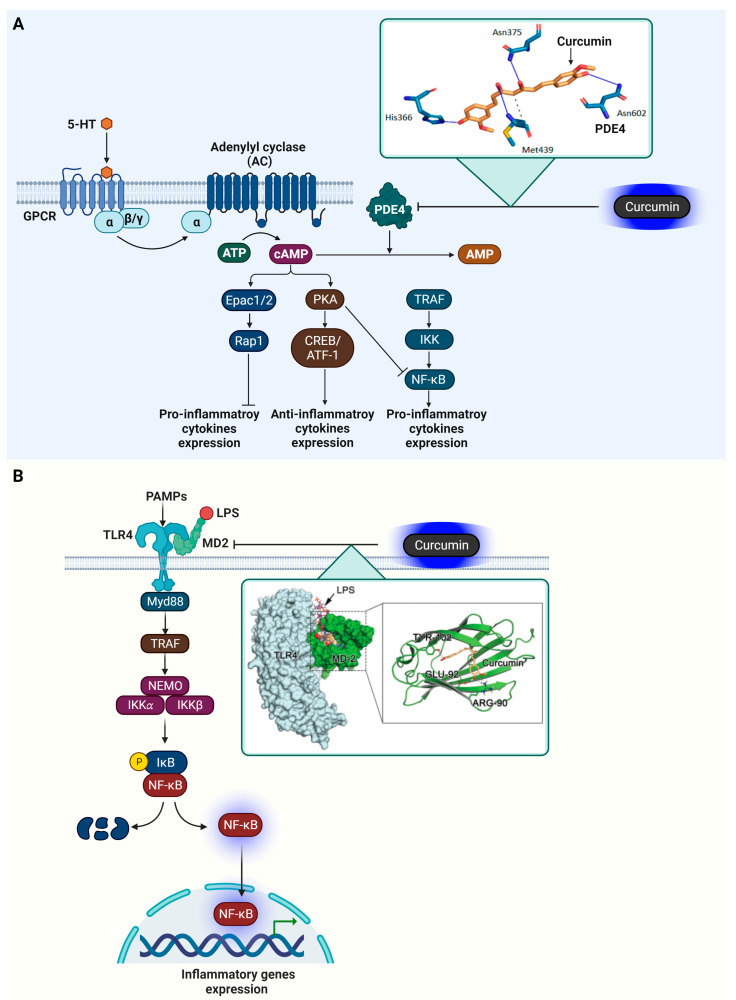
Curcumin’s regulatory role on PDE4 and MD-2 in inflammation. (**A**) PDE4 is involved in the hydrolysis of cAMP into AMP, affecting signaling through Epac1/2 and PKA, which play roles in cellular processes, including the upregulation of pro-inflammatory cytokines. Additionally, PDE4 modulation affects CREB/ATF-1 pathways, which are linked to anti-inflammatory cytokine production. An inset delineates the molecular interaction between curcumin and PDE4, showing curcumin forming hydrogen bonds with residues His336, Asn375, Met439, and Asn602, based on findings from docking studies [88]. (**B**) Curcumin’s interaction with MD-2 alters the conventional pro-inflammatory signaling initiated by PAMPs such as LPS. The binding of LPS to the TLR4/MD-2 complex leads to the recruitment of MyD88, activation of TRAF, and subsequent engagement of the NEMO complex with IKKα and IKKβ. This chain of events triggers IκB phosphorylation and degradation, liberating NF-κB to migrate to the nucleus and promote inflammatory gene expression. An inset magnifies curcumin’s binding to the TLR4/MD-2 complex, emphasizing the blockade of receptor activation via interaction with the amino acids Tyr102, Glu92, and Arg90, thus attenuating the inflammatory signaling, as identified in the docking study [141]. Green arrows indicate gene expression. The cartoon in Figure 7 was created with BioRender.com (https://app.biorender.com, accessed on 10 January 2024).

## 5. In Vitro Experiment Results of Curcumin Target Protein

In vitro studies play a pivotal role in elucidating the interactions of curcumin with various target proteins, offering crucial insights into its binding mechanisms and pharmacological impacts. These studies not only assess curcumin’s efficacy in influencing proteins integral to cancer and inflammatory pathways but also evaluate its safety and toxicity profile, paving the way for its application in clinical research. Furthermore, they shed light on curcumin’s bioavailability and stability, which is vital to the development of enhanced formulations. Thus, in vitro research is fundamental to advancing curcumin-based medical treatments.

Tae-Gyu Lim et al. revealed that curcumin markedly inhibits CDK2 kinase activity, induces G1 cell cycle arrest, and curbs cell proliferation in several colorectal cancer cell lines, including HCT116, HCT15, and DLD-1. The reduced effectiveness of curcumin in CDK2-knockdown cells underscores CDK2’s crucial role in curcumin’s anticancer efficacy [142].

Giorgio Cozza and colleagues demonstrated that curcumin inhibits CK2 kinase activity in a dose-dependent manner across various assays. However, its effectiveness declines in shorter incubations due to its conversion into ferulic acid. Ferulic acid inhibited CK2 effectively in cell lysates but had no significant impact on intact cells due to its low membrane permeability, in contrast to curcumin, which induced substantial cell death and effectively inhibited CK2 activity in Jurkat cells [36].

Shamim Akhtar Sufi and team found that the curcumin analogue ICA is more potent than curcumin in inhibiting GSK-3β kinase, inducing cell cycle arrest at both the G0/G1 and G2/M phases, and triggering apoptosis in the SW480 cell line [143].

Sourav Banerjee and colleagues discovered that curcumin specifically targets DYRK2, a regulator of the 26S proteasome, inhibiting its activity. This leads to reduced proteasome phosphorylation, decreased cancer cell proliferation, and enhanced apoptosis in combination with carfilzomib, with a reduced tumor burden observed in a breast cancer xenograft model [58].

Wei Li and team revealed that curcumin effectively suppresses the proliferation and invasive capabilities of pancreatic cancer cells induced by high glucose levels and EGF, primarily through the inhibition of the ERK and Akt pathways and by downregulating EGFR activation [144].

Kyung-Chan Kim and colleagues showed that curcumin effectively inhibits Axl phosphorylation upon Gas6 stimulation, leading to a significant decrease in cell viability, highlighting its potential as an anti-cancer agent targeting the Axl receptor [145].

Weiyong Hong and associates conducted a comprehensive study on curcumin-loaded nanoparticles, specifically targeting folate receptors in cancer cells. Their research included both in vitro and in vivo studies, demonstrating that these nanoparticles effectively reduce the viability of folate receptor-positive HeLa cancer cells in a dose-dependent manner and accumulate more efficiently in tumors than other organs in a HeLa xenograft mouse model, indicating successful and specific tumor targeting [146].

Miguel López-Lázaro et al. demonstrated that curcumin induces higher levels of Topo I and Topo II-DNA complexes in K562 leukemia cells than standard inhibitors, suggesting its potential as a unique cancer chemotherapeutic agent that operates via reactive oxygen species-mediated mechanisms [147].

Anlys Olivera et al. found that the curcumin analog EF31 is a more potent NF-κB inhibitor than EF24 or curcumin in RAW264.7 macrophages, showing a stronger inhibition of NF-κB DNA binding, nuclear translocation, and pro-inflammatory mediators, along with greater effectiveness in cancer cell lines and minimal toxicity in macrophages [148].

Huang et al. discovered that curcumin, applied topically, markedly reduces TPA-induced epidermal inflammation and arachidonic acid metabolism in mouse skin models, showcasing its strong anti-inflammatory properties without impacting protein kinase C activity in rat brain assays [149]. A summary of the in vitro experiment results for curcumin target proteins is displayed in Table 3.

## 6. Clinical Studies of Curcumin

Clinical research is essential for curcumin due to its promising therapeutic potential and diverse pharmacological properties, which necessitate rigorous testing to ascertain their efficacy and safety in humans. Furthermore, clinical studies are crucial to overcome curcumin’s bioavailability challenges and to develop effective dosage forms for its application in treating various health conditions.

### 6.1. Colorectal Cancer

Clinical investigations into curcumin’s ability to prevent colorectal cancer have been conducted. In a small trial involving five familial adenomatous polyposis (FAP) patients, a regimen of 480 mg curcumin and 20 mg quercetin thrice daily for six months led to a reduction in polyp size and number in four patients at three months, continuing for the same number at six months, with one patient dropping out after three months [150]. Another controlled trial with 44 FAP patients (21 on curcumin, 23 on placebo) taking 1500 mg of curcumin twice daily showed no significant differences between the groups [151]. A different study used two dosages (2000 mg and 4000 mg) of curcuminoid powder daily for 30 days in 40 participants, aiming to decrease procarcinogenic factors in aberrant crypt foci [152]. Only the higher dose group showed a 40% reduction in these foci, correlating with a significant increase in plasma curcumin levels.

### 6.2. Myeloid Leukemia (CML)

Imatinib, widely used in chronic myeloid leukemia (CML) treatment, was the focus of a study involving 50 CML patients. These patients were categorized into two groups: one received only imatinib (400 mg twice daily), while the other was treated with both imatinib and turmeric powder (5 g three times a day) for six weeks. The study observed that patients who received the combination of imatinib and turmeric powder had higher serum nitric oxide (NO) levels compared to those who received only imatinib. Although NO levels decreased in both groups, this reduction was more substantial in the group receiving the combined treatment. The study suggests that turmeric powder may enhance the treatment of CML by aiding in the reduction of NO levels. This finding points to the potential of turmeric powder as an adjunct therapy in CML treatment [153].

### 6.3. Prostate Cancer

A Phase II clinical trial assessed the effectiveness of docetaxel and curcumin in treating 30 patients with chemotherapy-naive metastatic castration-resistant prostate cancer (CRPC). These patients, experiencing progressive CRPC and rising prostate-specific antigen (PSA) levels, were administered docetaxel/prednisone over six cycles, along with oral curcumin (6 mg daily). The trial found no significant correlation between PSA levels and objective response rate with the serum levels of chromogranin A and neuron-specific enolase. The study provided new insights into curcumin’s effectiveness in cancer treatment, highlighting its high response rate, good tolerability, and patient acceptability [154].

### 6.4. Head and Neck Cancer

A randomized trial on oral leukoplakia treatment involved 223 patients, with 111 receiving 3600 mg of a curcumin-containing product and 112 on placebo, twice daily for six months [155]. The curcumin group showed a higher rate of complete or partial responses compared to placebo. No significant differences were noted after extending treatment to 12 months, and histological responses did not differ significantly between the groups.

### 6.5. Multiple Myeloma

The effect of curcumin on patients with monoclonal gammopathy of undetermined significance (MGUS) or smoldering multiple myeloma (SMM) was explored in a double-blind, placebo-controlled crossover study, followed by an open-label extension. Thirty-six patients (19 MGUS, 17 SMM) received 4000 mg of a curcumin-containing product or placebo for three months, then crossed over, with an option to continue at 8000 mg in an extension study [156]. The curcumin group showed decreases in free light-chain ratio and urinary markers of bone resorption, although no significant changes were noted in the placebo group post-crossover. The clinical results for curcumin are presented in Table 4.

## 7. Conclusions

Curcumin, a polyphenolic compound, is extracted from the turmeric plant. Its molecular architecture consists of two aromatic rings with ortho-methoxy phenolic groups, linked together by a seven-carbon chain featuring an α,β-unsaturated β-diketone component [157]. The significance of curcumin’s structure, especially its conjugated enol-ketone system, lies in its ability to engage in diverse molecular interactions like hydrogen bonding, hydrophobic contacts, and π–π stacking with different proteins. Proteins that curcumin interacts with, including CDK2, CK2α, GSK-3β, DYRK2, EGFR, AXL receptor, FR-β, DHFR, Topo I and II, and NF-κB, typically have specialized active or binding sites that can accommodate curcumin’s distinctive structure. These sites are often characterized by areas conducive to hydrophobic interactions, ATP-binding regions, and zones suitable for hydrogen and π–π bonding. Crucial amino acids in these proteins, such as lysine, glutamic acid, leucine, valine, and aspartic acid, are instrumental in forming these interactions with curcumin, predominantly via hydrogen bonds and hydrophobic interactions.

To enhance curcumin’s anticancer properties, structural alterations could focus on improving its stability, bioavailability, and protein-targeting precision. By increasing curcumin’s hydrophobic nature, its interactions with the hydrophobic regions of proteins could be strengthened. Adding functional groups that form robust hydrogen bonds or enhanced π–π interactions might increase its affinity for specific protein targets. Alterations to the linker chain or the aromatic ring could facilitate more stable and specific engagements with key protein residues. Customizing these modifications for precise protein targeting could lead to the development of more effective and selective anticancer compounds derived from curcumin. This understanding of curcumin’s structure and its ability to interact with a range of cancer-related proteins lays the groundwork for the development of modified curcumin derivatives with improved therapeutic potential in cancer treatment.

Based on the clinical trial results, the efficacy of curcumin in cancer therapy presents a complex scenario with both promising and inconclusive outcomes. In colorectal cancer, early studies indicated potential benefits of curcumin in reducing polyp number and size in familial adenomatous polyposis (FAP) patients, suggesting its role in managing precancerous lesions [150]. However, subsequent trials provided mixed results, with one study showing no significant difference between curcumin and placebo groups [151], while another observed a reduction in aberrant crypt foci at higher doses [152]. These findings suggest that while curcumin may have some preventive effects against colorectal cancer, its efficacy might be dose-dependent and variable across different patient populations.

For head and neck cancer, particularly oral leukoplakia, curcumin showed a significant clinical response compared to placebo [155]. However, the long-term benefits were unclear, as no significant difference was observed between the treatment arms after 12 months. This indicates term efficacy requires further investigation. It is also noteworthy that the combination of clinical and histological responses indicated a more pronounced effect of curcumin, suggesting its potential utility in the early-stage management of oral leukoplakia.

In the context of multiple myeloma, particularly in patients with monoclonal gammopathy of undetermined significance (MGUS) or smoldering multiple myeloma (SMM), curcumin demonstrated some promise. The study showed a decrease in serum markers associated with bone resorption and light-chain ratios in patients taking curcumin, although these changes were not statistically significant in all parameters [156]. The results from the crossover study followed by an open-label extension study suggest that curcumin may have a modulating effect on disease markers in MGUS and SMM, but its clinical significance in terms of disease progression and symptom management remains to be fully established.

In conclusion, while curcumin has shown some potential benefits in cancer therapy, its efficacy appears to be variable, dependent on factors such as dosage, cancer type, and individual patient response. These clinical trials underscore the need for more comprehensive and large-scale studies to fully elucidate the therapeutic potential of curcumin in cancer treatment. The promising results in certain areas warrant continued research to optimize curcumin’s formulation, delivery, and dosing to maximize its clinical benefits.

## Figures and Tables

**Table 1 ijms-25-02911-t001:** Target protein of curcumin in cancer.

Functional Classification	Target Protein	ProteinFunction	Compound	Binding Strength	Interactions Residues	Ref.
Kinase	CDK2	G_1_-S phage transition	Curcumin 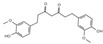	Binding affinity ^a^= −7.8 kcal/mol	Hydogen bond (Glu12, Leu83),Van der Waals(Val18, Ile10)	[21]
			Kurkumod23(curcumin derivative) 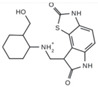	Binding affinity= −9.15 kcal/mol	Hydogen bond (Lys33, LEU83, Glu81),Van der Waals(Ala31, Ala44, Leu134, Val18, Phe80, Val64)	
Kinase	CK2α	Cancer cell proliferation	Ferulic Acid(curcumin derivative) 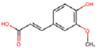	LE score ^b^= 0.61 kcal/mol	Hydogen bond (Lys68, Val116, Asp175)	[36]
Kinase	GSK3β	Involved Wnt and insulin signaling.	Curcumin	IC_50_ ^c^= 66.3 nM	Hydogen bond (Val135, Ile62)	[49]
Kinase	DYRK2	Cancer development and progression	Curcumin	IC_50_ = 5 nM	Hydogen bond (Lys251, Glu266, Asp368)Hydrophobic interaction(Ile228, Ala249, Ile285, Phe301, Leu303, Leu355, Ile367)	[58]
Receptor	EGFR	Cancer cell proliferation	3a(curcumin analog) 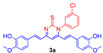	Docking ^d^ Score= −6.593 kcal/mol	π–π stacking (Asp855, Asp800, Leu718)	[66]
Receptor	AXL Tyrosine Kinase receptor	Promotes cancer development	CID 21159180(curcumin analog) 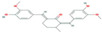	Binding affinity= −9.0 kcal/mol	Hydogen bond (Met623, Glu585, Asp627)	[82]
Receptor	FR-β	Folic acid uptake	Curcumin	Docking Score= −63.30 kcal/mol	Hydogen bond(Arg152, His151, Ser190, Asp97)π–π stacking ^f^(Trp187, Tyr101)	[88]
Enzyme	DHFR	DNA and RNA synthesis	Curcumin	ΔG ^e^ = −9.02 kcal/molKi = 243 nM	π–π stacking (Phe34)Hydogen bond(Glu30, Phe31, Val115, Tyr121)	[94]
Enzyme	Topo I and II	Relaxing DNA supercoils	Cyclo-Curcumin(curcumin derivative) 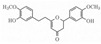	Topo IΔG = −10.33 kcal/molTopo IIΔG = −11.16 kcal/mol	Topo I(Asp479, Ser480, and Gln778)Topo II(Asp479, Ser480, Gln778)	[101]
Transcriptionfactor	NF-κB	Immune response, inflammation, and cell survival	Curcumin	Vina Score= −8.0 kcal/mol	Thr256, Ala257, Pro324, Pro344, Phe345, and Leu346 of p50	[108]

^a^ Binding Affinity—measures the strength of the interaction between a ligand and its target, with lower values indicating stronger interactions. ^b^ Ligand efficiency (LE) score—assesses the efficiency of a ligand in binding to a target relative to its size, with higher scores indicating better efficiency. ^c^ IC50—represents the concentration of a substance required to inhibit a biological process by 50%, used to evaluate the potency of inhibitors. ^d^ Docking score—a computational estimate of how well a ligand fits into the binding site of a target protein, with lower scores suggesting a better fit. ^e^ ΔG binding score: quantifies the change in energy when a ligand binds to a target, indicating the stability of the complex, with more negative values signifying stronger and more stable interactions. ^f^ π–π stacking—a non-covalent interaction between aromatic rings.

**Table 2 ijms-25-02911-t002:** Target protein of curcumin in inflammation.

Functional Classification	Target Protein	ProteinFunction	Compound	Binding Strength	Interactions Residues	Ref.
Enzyme	COX-2	Prostaglandin endoperoxide synthase	Curcumin	Binding affinity= −5.71 kcal/mol	Hydogen bond(Ala562)	[126]
			CDF (curcumin analog) 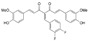	Binding affinity = −7.91 kcal/mol	Hydogen bond(Glu346, Phe580, Asn101, Gln350)	
Enzyme	CRP	Activate the complement system	Curcumin	Docking Score= −18.0033 kcal/mol	Hydogen bond(Asp140, Gln150)	[130]
Enzyme	PDE4	Modulation of cAMP signaling	Curcumin	Docking Score= −62.24 kcal/mol	Hydogen bond(His336, Asn375, Met439, Asn602)Hydrophobic interaction(Met439)	[88]
Co-factor protein	MD2	Component ofthe TLR4	Curcumin	K_D_ value= 0.000379 M	Hydogen bond(Arg90, Glu92, Tyr102)	[141]

**Table 3 ijms-25-02911-t003:** In vitro efficacy of curcumin.

Cell Line	Dose	Target Protein	Findings	Ref.
HCT116	5, 10, 20, 40 µM curcumin	CDK2	Curcumin suppresses CDK2 kinase activity dose-dependently and substantially reduces proliferation of these colon cancer cell lines. Induces G_1_ cell cycle arrest.	[142]
Jurkat Cells	10 µM curcumin	CK2 alpha	Curcumin induces significant cell death and inhibits CK2. Activity in Jurkat cells; comparable effects to the CK2 inhibitor CX-4945.	[36]
SW480	10 µM ICA (Curcumin analogue)	GSK3β	Inhibits GSK-3β kinase. Cell cycle arrest at G_0_/G_1_ and G_2_/M phase. Induces apoptosis.	[143]
MDA-MB-231 and HaCaT cells	10 μM curcumin	DYRK2	Curcumin decreases 26S proteasome activity by 25–40%.	[58]
Bortezomib-Resistant RPMI8226.BR and MM.1S.BR	10 μM curcumin	DYRK2	Comparable cytotoxicity to curcumin despite resistance to bortezomib.	
BxPC-3 (Pancreatic Cancer Cells)	20 μM curcumin	EGFR	Curcumin inhibited proliferation under high-glucose conditions and EGF stimulation. The invasive ability was suppressed by curcumin. Curcumin downregulated the activation of EGF/ERK and EGF/Akt pathways in high-glucose conditions and EGF treatment.	[144]
A549, H460	curcumin doses (up to 20 μM)	AXL Tyrosine Kinase receptor	Curcumin inhibited Gas6-induced Axl phosphorylation, suggesting an inhibitory effect on Axl activation. Treatment with curcumin reduced cell viability in a dose-dependent manner. At 20 μM, only 30% (A549) and 22% (H460) of cells survived, indicating significant anti-proliferative effects.	[145]
HeLa	2.5–40 μg/mL	FR-β	Curcumin-loaded nanoparticles showed the strongest anticancer activity, with an IC_50_ of 13.88 μg/mL. Enhanced cytotoxicity due to FR-mediated endocytosis.	[146]
K562 Leukemia	10 µM curcumin	Topo I and Topo II	Curcumin induced higher levels of Topo I and Topo II–DNA complexes than standard inhibitors.	[147]
RAW264.7 Macrophages	1–100 µM EF31 (Curcumin analog)	NF-κB	EF31 showed significantly more potent inhibition of LPS-induced NF-κB DNA binding than EF24 and curcumin, with an IC_50_ of ~5µM for EF31.	[148]
Mouse Epidermis	3, 10, 30, 100 µM	COX-2	Curcumin inhibited the metabolism of arachidonic acid to 5-HETE (40–83% inhibition) and 8-HETE (40–85% inhibition). IC_50_ = 5–10 µM. Curcumin markedly inhibited TPA- and arachidonic acid-induced epidermal inflammation.	[149]

**Table 4 ijms-25-02911-t004:** A concise overview of the research findings regarding curcumin’s effects on different types of cancer.

Cancer Type	Description	Ref.
Colorectal cancer	Trials on familial adenomatous polyposis (FAP) patients with doses of 480 mg curcumin and 20 mg quercetin, showing a reduction in polyp size and number in most patients. Another trial with 1500 mg curcumin twice daily showed no significant differences.	[151,152]
Myeloid leukemia (CML)	Combined treatment of turmeric powder and imatinib more effective in reducing NO levels, suggesting turmeric’s potential as an adjunct therapy in CML	[153]
Prostate cancer	The Phase II clinical trial concluded that the combination of docetaxel and curcumin in treating metastatic castration-resistant prostate cancer demonstrated a high response rate, good tolerability, and patient acceptability.	[154]
Head and neck cancer	Randomized trial on oral leukoplakia with 3600 mg curcumin-containing product twice daily for six months. A higher rate of complete or partial responses was seen compared to placebo, with no significant differences in extended treatment or histological responses.	[155]
Multiple myeloma	Double-blind, placebo-controlled crossover study on MGUS or SMM patients with 4000 mg curcumin-containing product. Decreases noted in free light-chain ratio and urinary markers of bone resorption.	[156]

## Data Availability

Not applicable.

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
