# Peer review of "Curcumin in Cancer and Inflammation: An In-Depth Exploration of Molecular Interactions, Therapeutic Potentials, and the Role in Disease Management"

_ijms, 2024, doi:10.3390/ijms25052911_

Round 1
Reviewer 1 Report
Comments and Suggestions for Authors This paper reviews the target proteins of curcumin based on the data of binding strength. Since multilple mechanisms have been proposed for the anti-cancer effects of curcumin so far, this manuscript covers relevant topics in this field. This reviewer has following major comments.- In Table 1 and 2, since the binding strength is expressed using different units (e.g., binding affinity, LE score, IC50, docking score), it is difficult to understand which protein has a higher binding strength to curcumin. It will be helpful for readers if authors use the same unit as much as possible and sort proteins by binding strength (i.e., the protein that is likely to have the highest binding strength is shown at the top).
- In the text, it is recommended to introduce proteins and pathways in the order of the binding strength of curcumin.
- ROS production by curcumin has been published by several independent research groups. However, it is not mentioned in this manuscript.
- Excluding references, the manuscript is 35 pages long and takes time to read.
Author Response
This paper reviews the target proteins of curcumin based on the data of binding strength. Since multilple mechanisms have been proposed for the anti-cancer effects of curcumin so far, this manuscript covers relevant topics in this field. This reviewer has following major comments.
- In Table 1 and 2, since the binding strength is expressed using different units (e.g., binding affinity, LE score, IC50, docking score), it is difficult to understand which protein has a higher binding strength to curcumin. It will be helpful for readers if authors use the same unit as much as possible and sort proteins by binding strength (i.e., the protein that is likely to have the highest binding strength is shown at the top).
⟶ Thank you for your valuable feedback regarding the presentation of binding strength data in Tables 1 and 2. However, the use of different units (e.g., binding affinity, LE score, IC50, docking score) for expressing binding strength is based on referenced literature, making it challenging to unify them into a single unit. Each metric measures distinct aspects of molecular interaction, making direct comparison difficult without conversion or normalization. Binding affinity quantifies ligand-receptor interaction strength. LE score relates binding affinity to ligand size. IC50 measures concentration for 50% activity inhibition, and docking scores estimate binding affinity computationally. Thus, while I acknowledge the complexity this presents, the diversity of units is essential for accurately conveying the nuances of each interaction as per the referenced literature.
- In the text, it is recommended to introduce proteins and pathways in the order of the binding strength of curcumin.
- ROS production by curcumin has been published by several independent research groups. However, it is not mentioned in this manuscript.
⟶ Curcumin induces ROS production through multiple pathways, affecting mitochondrial function and electron transport, impacting NADPH oxidases, and modulating antioxidant enzymes. This increases ROS levels, contributing to curcumin's anti-inflammatory and anticancer effects by initiating cell responses leading to cancer cell death or inflammation modulation. However, our review focuses on proteins binding to curcumin, and due to a lack of studies specifically addressing proteins that impact ROS production in relation to curcumin binding, this topic was not included.
- Excluding references, the manuscript is 35 pages long and takes time to read.
⟶ I acknowledge the concern about the manuscript's length and its impact on readers. To address this, I have thoroughly reviewed the document to remove redundancies, highlight key findings, and streamline discussions, aiming for brevity and enhanced readability.
Reviewer 2 Report
Comments and Suggestions for Authors
This is a very well-written paper, I am impressed by the effort the authors put into its preparation. I think that once it is published, I will be able to point it out to my PhD students as an example of a well-written work. In addition to the text, the figures, which are very easy to understand, deserve attention. However, I would like to ask whether they were not prepared using the Biorender service, if so, it should be cited. Moreover the meaning of the yellow circles with "P" should be explained. Another thing that would further improve this article is a figure that would present the analogs of curcumin mentioned in the article. When listing the curcumin analogs, the authors write about the differences in their action and effect, but the reader may not know what the chemical formulas of analogs such as EF31 line 915; CDF line 712; CID21159180 line 409; analog 3a line 354; cyclocurcumin (Tab1) and others look like. Moreover, these compounds are called analogs in some places of the manuscript and as derivatives in others, while analog and derivative are not the same. Analogue: In the context of chemistry, an analogue refers to a chemical compound that has a structure similar to another compound, but differs in some components or functional groups. A derivative is a chemical compound that is formed by modifying one or more molecular components of another compound (called the parent compound).
Author Response
This is a very well-written paper, I am impressed by the effort the authors put into its preparation. I think that once it is published, I will be able to point it out to my PhD students as an example of a well-written work. In addition to the text, the figures, which are very easy to understand, deserve attention. However, I would like to ask whether they were not prepared using the Biorender service, if so, it should be cited.
⟶ All figure legends have been annotated to indicate they were created using Biorender.
Moreover the meaning of the yellow circles with "P" should be explained.
⟶ The following sentence has been added: 'Yellow circles with 'P' represent phosphorylation.'
Another thing that would further improve this article is a figure that would present the analogs of curcumin mentioned in the article. When listing the curcumin analogs, the authors write about the differences in their action and effect, but the reader may not know what the chemical formulas of analogs such as EF31 line 915; CDF line 712; CID21159180 line 409; analog 3a line 354; cyclocurcumin (Tab1) and others look like. Moreover, these compounds are called analogs in some places of the manuscript and as derivatives in others, while analog and derivative are not the same. Analogue: In the context of chemistry, an analogue refers to a chemical compound that has a structure similar to another compound, but differs in some components or functional groups. A derivative is a chemical compound that is formed by modifying one or more molecular components of another compound (called the parent compound).
⟶ Thank you for your recommendation to include a figure displaying the chemical structures of the curcumin analogs and derivatives mentioned in our manuscript. Recognizing the value of visual aids for understanding the chemical variations and their biological implications, I have added a figure depicting the structures of EF31, CDF, CID21159180, analog 3a, cyclocurcumin, and others as discussed. Additionally, I have taken note of your observation regarding the inconsistent terminology of 'analogs' and 'derivatives.' I have corrected the incorrectly marked sections.